# Temporal analysis of enhancers during mouse cerebellar development reveals dynamic and novel regulatory functions

**Miguel Ramirez[1,2], Yuliya Badayeva[1,2], Joanna Yeung[1,2], Joshua Wu[1,2], Ayasha Abdalla-Wyse[1,2], Erin Yang[1,2], FANTOM 5 Consortium[3], Brett Trost[4], Stephen W Scherer[4], Daniel Goldowitz[1,2]***

[1]Centre for Molecular Medicine and Therapeutics, BC Children's Hospital Research Institute, Vancouver, Canada; [2]University of British Columbia, Vancouver, Canada; [3]RIKEN, Wako, Japan; [4]The Centre for Applied Genomics, The Hospital for Sick Children, Toronto, Canada

**\*For correspondence:**
dang@cmmt.ubc.ca

**Abstract** We have identified active enhancers in the mouse cerebellum at embryonic and post-natal stages which provides a view of novel enhancers active during cerebellar development. The majority of cerebellar enhancers have dynamic activity between embryonic and postnatal development. Cerebellar enhancers were enriched for neural transcription factor binding sites with temporally specific expression. Putative gene targets displayed spatially restricted expression patterns, indicating cell-type specific expression regulation. Functional analysis of target genes indicated that enhancers regulate processes spanning several developmental epochs such as specification, differentiation and maturation. We use these analyses to discover one novel regulator and one novel marker of cerebellar development: Bhlhe22 and Pax3, respectively. We identified an enrichment of de novo mutations and variants associated with autism spectrum disorder in cerebellar enhancers. Furthermore, by comparing our data with relevant brain development ENCODE histone profiles and cerebellar single-cell datasets we have been able to generalize and expand on the presented analyses, respectively. We have made the results of our analyses available online in the Developing Mouse Cerebellum Enhancer Atlas, where our dataset can be efficiently queried, curated and exported by the scientific community to facilitate future research efforts. Our study provides a valuable resource for studying the dynamics of gene expression regulation by enhancers in the developing cerebellum and delivers a rich dataset of novel gene-enhancer associations providing a basis for future in-depth studies in the cerebellum.

## Editor's evaluation

This manuscript is a valuable study to understand regulatory elements in the early developing mouse cerebellum. The authors have generated important ChIP-seq datasets from embryonic and early postnatal mouse cerebellum. The authors use these data to convincingly identify enhancers in the developing cerebellum. The authors have also made their data and analyses available online in the "Developing Mouse Cerebellum Enhancer Atlas."

## Introduction

Neuronal development is a complex and dynamic process that involves the coordinated generation and maturation of countless cell types. For cerebellar granule cells, the most numerous neurons in the brain, neuronal differentiation consists of several steps beginning with the commitment of neural

stem cells to become specified neural precursors, followed by multiple migratory stages to reach and mature at its final destination (*Consalez et al., 2020*). Underpinning these events is the expression of gene regulatory networks that drive dynamic molecular processes required for proper brain formation (*Ziats et al., 2015*). However, the transcriptional mechanisms that precisely regulate these gene expression programs have not been fully described.

Gene expression is typically activated when transcription factors (TFs) bind to non-coding regulatory elements and recruit the necessary components to begin transcription. Among the several classes of non-coding sequences that regulate gene expression, enhancers are the most common, with thousands predicted to coordinate transcriptional regulation during development (*Heinz et al., 2015*). Enhancers are regulatory sequences that serve as binding sites for TFs and activate distal target gene expression. In the brain, enhancers help to ensure that gene expression is spatially- and temporally-specific, defining what genes will be active during distinct stages of development (*Nord and West, 2020*). Transcriptional regulation by enhancers has been shown to be critical for cellular identity, maturation during central nervous system (CNS) development, and activity-dependent responses in mature neurons (*Frank et al., 2015*; *Pattabiraman et al., 2014*). A detailed understanding of the enhancers that govern changes in gene expression during embryonic and early postnatal cerebellum development remains limited. Profiling genome-wide enhancer activity at different time points and identifying their gene regulatory targets can provide insight into developmental processes regulated by enhancer elements.

Several molecular properties have been associated with enhancer activity, and the advancement of sequencing technology has facilitated their identification genome-wide in several developing brain structures (*Carullo and Day, 2019*). Enhancers are marked with histone post-translational modifications H3K4me1 and H3K27ac, both of which contribute to opening chromatin for TF binding (*Calo and Wysocka, 2013*). H3K27ac delineates active from poised elements and has been a reliable marker for enhancer activity genome-wide (*Creyghton et al., 2010*). Analysis of these marks, in conjunction with transcriptomic and epigenomic datasets, has revealed that the vast majority of non-coding variants associated with neurological and psychiatric disorders are found within these regulatory elements, highlighting their importance in functional readout in the brain (*Barešić et al., 2020*). Thus, profiling enhancer-associated histone modifications in the brain across time provides a comprehensive understanding of gene-regulatory principles, disease-associated variants, and the genetics of brain development (*Nott et al., 2019*).

The cerebellum has been a long-standing model to study the developmental genetics of the brain. This is, in part, due to the limited number of cell types, well-defined epochs of development for these cell types and a simple trilaminar structure in which these cells are organized, making for an enhanced resolution of events in time and space (*Wang and Zoghbi, 2001*). Our lab has previously generated transcriptomic time course datasets spanning embryonic and postnatal development using microarrays and Cap Analysis of Gene Expression followed by sequencing (CAGE-seq; *Forrest et al., 2014*; *Ha et al., 2019*), which led to the discovery of novel TFs critical for proper development. More recently, the developing cerebellum has served as an ideal setting for pioneering mouse and human single-cell RNA-seq time courses (*Aldinger et al., 2021*; *Carter et al., 2018*; *Peng et al., 2019*; *Wizeman et al., 2019*). These studies have provided an unbiased classification of cerebellar populations using their associated gene expression profiles and insight into the cellular origins of distinct postmitotic cell subtypes from neural progenitors.

When considering the non-coding regulatory elements, such as enhancers, that fine tune the spatial and temporal expression of these genes, there are only a few studies that have profiled these sequences genome-wide and specific to the developing cerebellum. Enhancer-associated histone marks (H3K27ac and H3K4me1) have been identified in postnatal and adult mouse cerebella (*Frank et al., 2015*). During embryonic development, these marks have been profiled in the mouse hindbrain, but not specifically in the cerebellum (*Gorkin et al., 2020*). Recently, the open chromatin landscape has been examined during embryonic and postnatal cerebellum development using single-nuclear ATAC-seq (snATAC-seq) (*Sarropoulos et al., 2021*). While this analysis has provided a comprehensive atlas of predicted regulatory sequences at cell-type resolution, additional signals associated with enhancer activity, such as H3K4me1 and H3K27ac, have yet to be quantified during embryonic and early postnatal development. Profiling these chromatin marks can provide further evidence of regulatory activity and is an important step in establishing a catalog of active enhancers involved in

cerebellar development. Identifying enhancers and predicting their target genes would also provide a valuable resource for the research community and would facilitate the discovery of novel genetic drivers of the precisely-timed and cell-specific molecular events in the developing cerebellum.

We utilize chromatin immunoprecipitation followed by sequencing (ChIP-seq) of enhancer associated histone marks H3K4me1 and H3K27ac at 3 stages of embryonic and early postnatal cerebellar development. We identify temporally specific enhancers using a differential peak analysis comparing postnatal and embryonic timepoints. TF motif enrichment and prediction of gene targets led to the elucidation of molecular processes regulated by enhancers during these stages. As examples of the use of this data for discovery, we identify two novel regulators of cerebellar development, Pax3 and Bhlhe22: a novel marker of GABAergic progenitors and a regulator of postnatal granule cell migration, respectively. Finally, we also demonstrate how this data may be used to explore the role of enhancers in neurodevelopmental disabilities by identifying an enrichment of autism spectrum disorder (ASD) associated SNPs and de novo variants found in ASD-affected individuals in cerebellar enhancers. We have made the results of our analyses available online in the Developing Mouse Cerebellum Enhancer Atlas which can be easily queried, curated and exported by the research community. This provides a rich resource that can be utilized to discover novel genetic regulators of cerebellar development

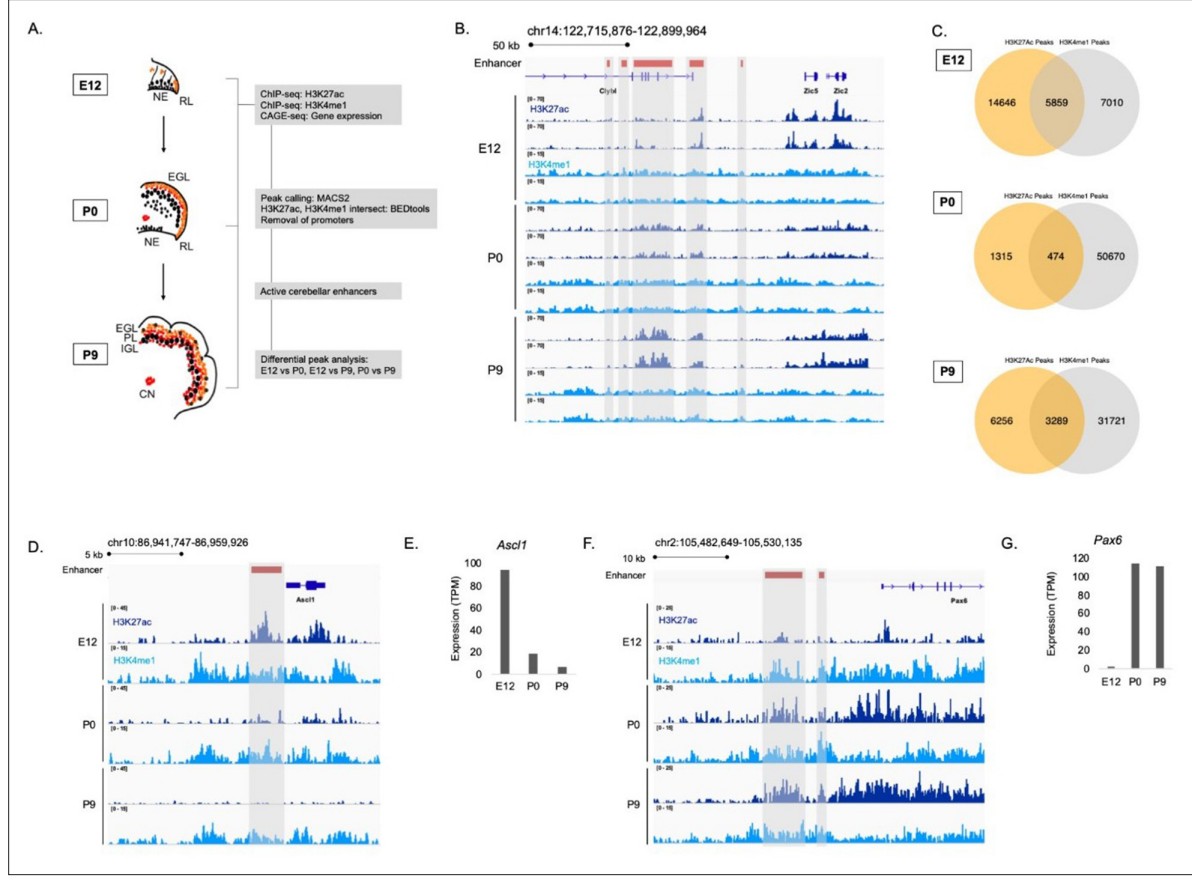

**Figure 1.** Enhancer identification during cerebellar development. (**A**) An overview of the stages of cerebellar development profiled in this study. The datasets collected at these ages and the downstream analyses are shown in the flow chart. Labels: NE: Neuroepithelium, RL: Rhombic lip, EGL: External granular layer, PL: Purkinje layer, IGL: Inner granular layer, CN: Cerebellar nuclei (**B**) A region of the mouse genome chr14:122,715,876–122,899,964 (mm9) in the Integrative Genomics Viewer (IGV) showing H3K27ac and H3K4me1 profiles across biological replicates of E12, P0, P9 cerebella. Active cerebellar enhancers are highlighted (gray box). (**C**) Venn diagrams displaying overlap between H3K27ac and H3K4me1 peaks at E12, P0 and P9. (**D–E**) An example of a cerebellar enhancer identified from the E12 cerebellum. Shown is normalized H3K27ac and H3K4me1 signal at the enhancer (gray box), as well as (**E**) normalized CAGE-seq expression of the nearest gene, *Ascl1*, across developmental time, at E12, P0, P9. TPM, Transcripts Per Million. (**F–G**) An example of a cerebellar enhancer identified from the P9 cerebellum. Shown is normalized H3K27ac and H3K4me1 signal at the enhancer (gray box), as well as (**G**) normalized (TPM) CAGE-seq expression of the nearest gene, *Pax6*, across developmental time, at E12, P0, P9.

The online version of this article includes the following figure supplement(s) for figure 1:

**Figure supplement 1.** Validation of identified enhancer sequences.

and contribute to understanding the impact of genetic variation associated with neurodevelopmental disorders.

## Results

### Enhancer identification during cerebellar development

To identify enhancers active during embryonic and postnatal cerebellar development, we generated genome-wide H3K27ac and H3K4me1 ChIP-seq profiles from mouse cerebella dissected at embryonic day 12 (E12), postnatal day 0 (P0) and postnatal day 9 (P9) (*Figure 1A*). These stages represent a window of time that spans important neurodevelopmental events such as cell specification, emergence from the cell cycle, differentiation, migration, and maturation for all of the major neuronal types of the cerebellum including the cerebellar granule cells, Purkinje cells, cerebellar interneurons, and cerebellar nuclear neurons (*Goldowitz and Hamre, 1998*; *Hatten and Heintz, 1995*; *Wang and Zoghbi, 2001*). H3K27ac and H3K4me1 signals were reproducible between biological replicates as exemplified in a region on chromosome 14 (*Figure 1B*). There was a high correlation between replicates for both marks at each age (*Figure 1—figure supplement 1A*). Therefore, we had confidence in using our H3K27ac and H3K4me1 data in downstream analyses. Robust cerebellar enhancers were identified by the presence of overlapping peaks between the two enhancer-associated histone marks at each age. This highlighted a total of 9,622 peaks; 5,859, 474, and 3,289 peaks that were in both the H3K27ac and H3K4me1 datasets at E12, P0, and P9, respectively (*Figure 1C*). Duplicate peaks between ages were removed, producing a list of **7024** active cerebellar enhancers derived from overlapping H3K27ac and H3K4me1 signals (*Supplementary file 1*).

The relationship between enhancer activity and genes relevant to cerebellar development is shown in genomic regions flanking *Ascl1* and *Pax6*, two genes critical to cerebellar development (*Kim et al., 2008*; *Yeung et al., 2016*). We identified an enhancer active at E12 located in close proximity to *Ascl1* (*Figure 1D*). A decrease in the H3K27ac ChIP-seq signal at this enhancer corresponded to a decrease in *Ascl1* gene expression (*Figure 1E*). We identified two active enhancers at P9 located near *Pax6* (*Figure 1F*). H3K27ac ChIP-seq signal also showed a pattern of activity similar to *Pax6* expression, increasing from embryonic to postnatal ages (*Figure 1G*). These results provide validation for the enhancers identified in our dataset relative to genes critical to cerebellar development.

We compared our list of robust cerebellar enhancers to four previously published cerebellar enhancer datasets. First, P7 H3K27ac ChIP-seq and DNase-seq profiles previously generated by *Frank et al., 2015* were overlapped with robust cerebellar enhancers. Greater than 90% of our reported robust cerebellar enhancers are replicated by H3K27ac and DNAse-seq peaks from this study (*Figure 1—figure supplement 1B-C*). Second, we compared the number of robust cerebellar enhancers and our histone profiles identified at P9 to H3K27ac and H3K4me1 ChIP-seq datasets from the adult cerebellum (*Gorkin et al., 2020*). We found that 73% (7037/9545) of H3K27ac peaks, 59% (20674/35010) H3K4me1peaks and 60% (1974/3289) of robust cerebellar enhancers overlapped with peaks identified in the adult cerebellum (*Figure 1—figure supplement 1D*). This indicates that a subset of the enhancers identified at P9 are active specifically during postnatal development when compared to adult stages. Third, enhancers retrieved from the enhancer database EnhancerAtlas 2.0, reporting enhancer activity in the mouse cerebellum at P0-P14 (*Gao and Qian, 2020*), were compared to robust cerebellar enhancers. We found that 73%, and 80% of our enhancers overlapped with the postnatal cerebellum enhancer dataset at P0, and P9, respectively (*Figure 1—figure supplement 1E*). Fourth, mouse enhancers that had experimentally validated hindbrain activity at E11.5 from the VISTA Enhancer Browser (*Visel et al., 2007*) were compared to the cerebellar enhancers reported here. We found that 56% of VISTA enhancers overlap with our cerebellar enhancer sequences at E12 (*Figure 1—figure supplement 1F*). Additionally, we found that a higher proportion of H3K27ac peaks overlapped with VISTA enhancers compared to open chromatin regions identified in the developing cerebellum (*Sarropoulos et al., 2021*) and the developing hindbrain (*Gorkin et al., 2020*; *Figure 1—figure supplement 1G*). This indicates that our identified H3K27ac peaks may be more predictive of active enhancers than previously generated chromatin accessibility datasets. Taken together, these confirmative findings indicate our approach was effective in capturing active cerebellar enhancers.

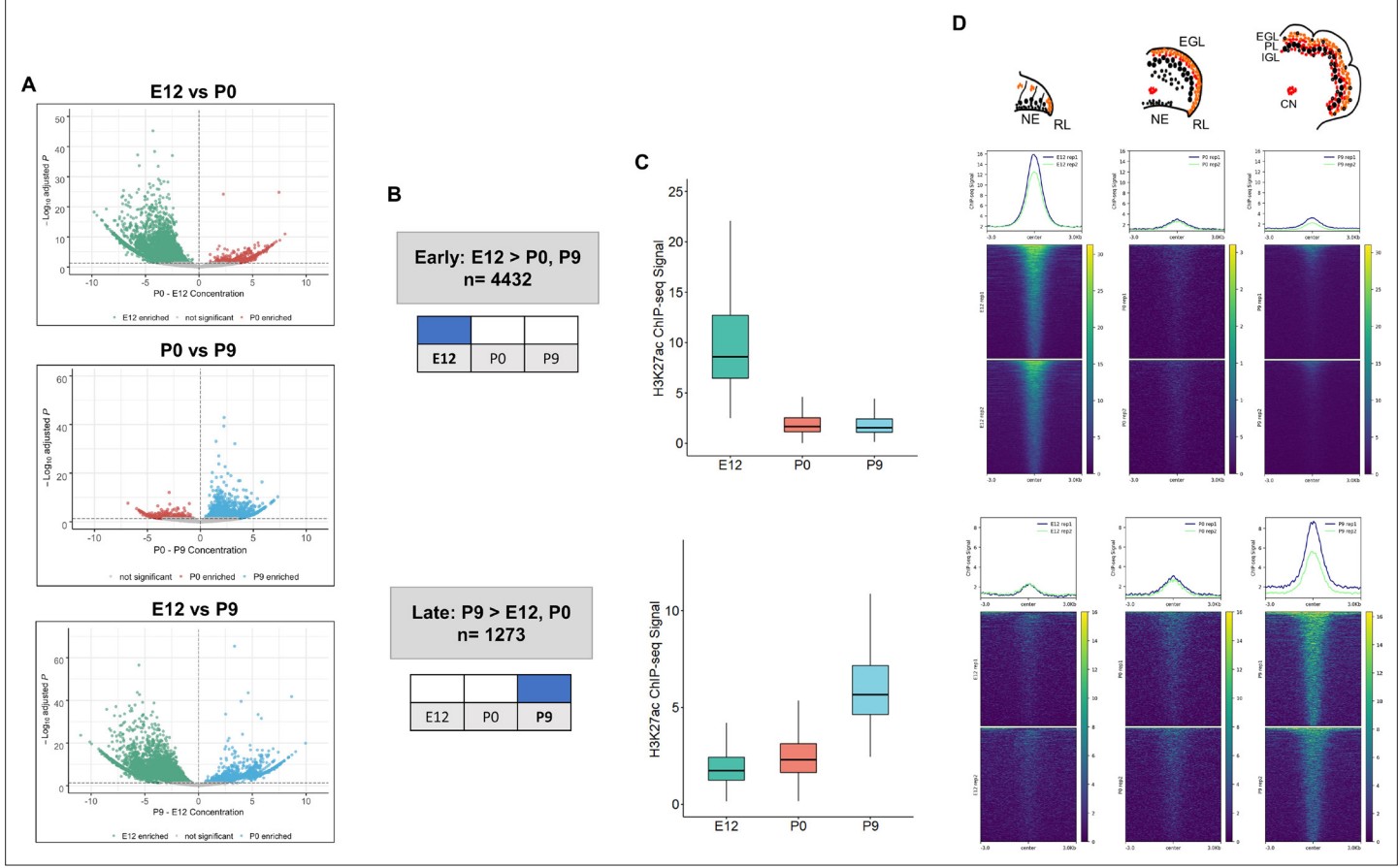

**Figure 2.** Enhancer activity is dynamic throughout cerebellar development. (**A**) Volcano plots showing robust cerebellar enhancers with differential H3K27ac peak signal for three comparisons: E12 vs P9, E12 vs P0, and P0 vs P9. Differential signal strength was identified for 4433 and 4355 robust cerebellar enhancers when comparing E12 to P9 and to P0, respectively. At P9, 1275 and 403 robust cerebellar enhancers had differential signal when compared to E12 and P0, respectively. Enhancers with significant differential activity are colored at a cutoff of an adjusted *P*-value <0.05. Displayed on the y-axis is the negative log10 adjusted p-value and on the x-axis is the difference in ChIP-seq signal between to the ages for a given peak. (**B**) Diagram displays how Early and Late active cerebellar enhancers are classified based on differential peak analysis results. (**C**) Boxplot shows mean ChIP-seq signal (y-axis) for all Early (upper) and Late (lower) active enhancers. Error bars represent the standard error of the mean. (**D**) Mean profile and heatmaps of H3K27ac signal at the midpoint of our predicted cerebellar enhancers (rows ±3 kb) in Early and Late groups at E12, P0, and P9.

The online version of this article includes the following figure supplement(s) for figure 2:

**Figure supplement 1.** Overlap with histone profiles from the developing hindbrain, midbrain and forebrain.

**Figure supplement 2.** Donut chart displaying the proportion of robust cerebellar enhancers that were found to overlap with one, two or three brain regions examined (hindbrain, midbrain and forebrain) or found to be active only in cerebellar samples (CB only).

**Figure supplement 3.** Box plots of Early and Late active enhancer activity for robust cerebellar enhancers with overlapping activity in the developing hindbrain, midbrain and forebrain.

**Figure supplement 4.** Cell type assignment of robust cerebellar enhancers using snATAC-seq as generated by *Sarropoulos et al., 2021* in the developing cerebellum.

## Enhancer dynamics during cerebellar development

The dynamics of enhancer activity over cerebellar development were examined through a differential peak analysis of H3K27ac signal. The majority, **89%** (**6238/7023**)**,** of cerebellar enhancers had significant differences in peak signal (adjusted *P*-value ≤0.05) throughout cerebellar development (*Figure 2A*, *Supplementary file 2*). At P9, **1273** cerebellar enhancers were significantly active compared to either P0, E12 or both. At E12, **4432** active enhancers were differentially active compared to either P9, P0 or both. At P0, in contrast, only a small number of enhancers with differential signal was identified (403 and 154 showed significant changes when compared to E12 and P9, respectively). However, none of these P0 cerebellar enhancers were differentially active when compared to both

E12 and P9, indicating that enhancer activity did not spike at birth. Quality control metrics evaluating sensitivity of the H3K27ac and H3K4me1 ChIPs (*Supplementary file 3*), such as the fraction of reads in peaks (FRiP) and relative strand correlation (RSC), all met ENCODE standards (*Landt et al., 2012*). The P0 samples had slightly lower scores, on average, in both metrics than the E12 and P9 samples, suggesting a minor decrease in sensitivity compared to the other stages. We acknowledge that the difference in sensitivity may have masked P0-specific enhancer activity in our analysis and that future experiments may reveal temporally-specific enhancers at this stage. Taken together, this analysis highlights two temporally specific windows of enhancer activity at Early (embryonic) and Late (postnatal) stages (*Figure 2B*).

Distinct patterns of enhancer activity were observed for temporally classified enhancers. For Early active enhancers, there was a loss of mean H3K27ac signal over time, with a steep decline after E12 (*Figure 2C*). Late active enhancers exhibited a gain in activity over time, with mean H3K27ac signal increasing steadily through development. These patterns are seen when looking at the changes in signal flanking the summits of our peaks across time (*Figure 2D*). These results indicate that the majority of cerebellar enhancers are dynamic throughout time and exhibit temporally specific activity.

## A subset of cerebellar enhancers is active in other developing regions of the brain

To examine cerebellar enhancer activity in the context of other developing brain regions, we overlapped our histone profiles and robust cerebellar enhancer coordinates with H3K4me1 and H3K27ac profiles previously generated from either the developing mouse hindbrain, midbrain or forebrain at E12 and P0 (*Gorkin et al., 2020*). At E12, we found that the majority of cerebellar H3K27ac and H3K4me1 peaks overlapped with profiles generated in either the hindbrain, midbrain, or forebrain (*Figure 2—figure supplement 1A*). Similar results were observed at P0, with the majority of cerebellar H3K27ac peaks identified in other brain regions; however, only a modest overlap was found for H3K4me1 profiles at this stage (*Figure 2—figure supplement 1B*). We found that most robust cerebellar enhancers overlapped with H3K27ac peaks detected in the hindbrain (89.1%), midbrain (78.6%), or forebrain (78.6%; *Figure 2—figure supplement 1C*). When looking at multiple brain regions, 69.8% of robust cerebellar enhancers were active in all three developing brain regions (*Figure 2—figure supplement 2*). Interestingly, the majority of the enhancers active in these brain regions (73.9%) were classified as Early active enhancers in the cerebellum which display temporally specific activity to embryonic development. Importantly, we identified 467 (6.7%) robust cerebellar enhancers specifically active in the cerebellum and not the other brain regions, indicating our analysis has uncovered novel enhancer sequences potentially critical for cerebellar formation (*Figure 2—figure supplement 2*). Over half of the cerebellar specific enhancers (55.03%) were classified as Late active enhancers with activity peaking during postnatal development, while only 15.92% of these enhancers were Early active enhancers and the remaining 29.05% of enhancers were not differentially active (labeled as 'Neither'). We then examined whether Early and Late and enhancers displayed similar patterns of activity in other brain regions from E12 to P0. Early enhancers displayed a decrease in average activity over time in the hindbrain, midbrain and forebrain (*Figure 2—figure supplement 3*). Late enhancers showed an increase in hindbrain samples, but minimal changes in the developing midbrain and forebrain (*Figure 2—figure supplement 3*). These collective results indicate that Early enhancers may also be active in other developing regions of the brain while Late active enhancers may be more likely to be active specifically in the cerebellum.

## Comparison of robust cerebellar enhancers with chromatin accessibility identified in single cells

To gain a more granular view of the spatial activity of robust cerebellar enhancers, we overlapped their coordinates with cis-regulatory elements (CREs) active in the developing cerebellum previously identified by single-nuclei ATAC-seq (snATAC-seq; *Sarropoulos et al., 2021*). We found that **6342/7024** (**90.1%**) robust cerebellar enhancers overlapped with CREs with open chromatin conformation. Cell-types were then assigned to robust cerebellar enhancers based on the activity of an overlapping CRE. CREs were previously aggregated into 26 clusters based on activity using an iterative clustering procedure (*Sarropoulos et al., 2021*). Using these clusters, we were able to assign a predicted cell type to the **6342** robust cerebellar enhancers that overlapped with CREs (*Figure 2—figure supplement*

**4A**). The predicted cell types with the largest proportion of robust cerebellar enhancers were granule cells (19.32%), progenitor cells (18.35%), Purkinje cells (11.48%) and multiple early-born neuron types (9.69%; *Figure 2—figure supplement 4A*). When splitting cerebellar enhancers into Early and Late active groups, we found that the majority of Early enhancers were predicted to be active in progenitor cells, and multiple early born neuron types such as Purkinje cell precursors while Late enhancers were predicted to be active in developing granule cells, which is the dominant population of cells produced during postnatal stages, and other late-born cell types (*Figure 2—figure supplement 4B*).

We then examined the average open chromatin signal at our established Early and Late robust cerebellar enhancers in progenitor cells and the predominant neuron types: granule cells, Purkinje cells, and interneurons. Early enhancers peaked in accessibility at embryonic stages and steadily declined throughout developmental time for all cell types (*Figure 2—figure supplement 4C*). Late enhancers gradually increased in accessibility throughout time peaking at postnatal stages for progenitor cells, granule cells and interneurons. In Purkinje cells, Late active enhancers showed a minimal change in accessibility throughout time (*Figure 2—figure supplement 4C*). These findings indicate that Early and Late enhancers display their respective activity patterns in the context of chromatin accessibility in individual cell-types.

## Cerebellar enhancers are enriched for neural transcription factor binding sites in an age-dependent manner

We then sought to identify transcription factors whose activity is dictated by the availability of robust cerebellar enhancers, as many neural lineage-defining factors drive cell commitment in the developing brain through enhancer binding (*Elsen et al., 2018*; *Lindtner et al., 2019*). We used HOMER to search for enriched motifs (adjusted p-value <1E-11) in Late and Early active cerebellar enhancers; which were then matched to known transcription factor motifs in the JASPAR database (*Heinz et al.,*

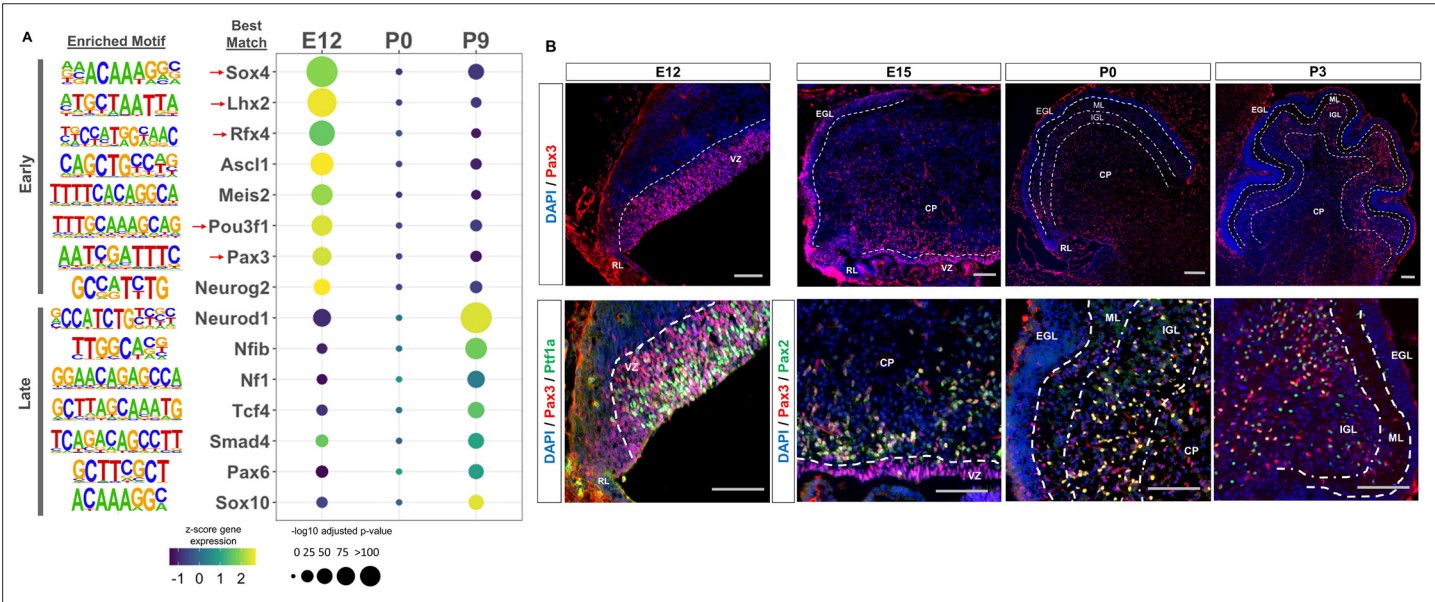

**Figure 3.** Neural transcription factors with known and novel function in the developing cerebellum are enriched in dynamic cerebellar enhancers. (**A**) Dot plot displaying significantly enriched (adjusted p-value <1E-11) motifs and the predicted matching transcription factor (TF). Displayed are the results for Early (top) and Late (bottom) active enhancers. TFs with an unknown functional role in cerebellar development are indicated with a red arrow. Size of the dots indicate the negative log10 adjusted p-value for a given motif and the color scale displays the z-score normalized expression throughout the cerebellar developmental time course. (**B**) Top: Immunofluorescent staining of Pax3 in the mouse cerebellum at E12, E15, P0, and P3. Bottom: Pax3 and Ptf1a immunofluorescent co-staining of the E12 mouse cerebellum. Immunofluorescent co-staining of Pax3 and Pax2 in the mouse cerebellum at E15 and P0. Labels: CP: Cerebellar parenchyma, EGL: External granular layer, NGL: Nascent granular layer, RL: Rhombic lip, VZ: Ventricular zone, Scalebars = 100 μm.

The online version of this article includes the following figure supplement(s) for figure 3:

**Figure supplement 1.** Immunofluorescent analysis of Pax3 expression in the developing mouse cerebellum.

**Figure supplement 2.** Pax3 expression in the developing mouse and human cerebellum quantified by CAGE-seq and scRNA-seq.

*2010*). This analysis revealed a distinct set of significantly enriched motifs for Early and Late enhancers (*Figure 3A*). The TFs with the best matching DNA binding motifs belonged to protein families with known regulatory roles in cerebellar development, serving as validation for our analysis (*Figure 3A*). Since TF motifs are similar between protein family members, it is possible that Early and Late cerebellar enhancers are bound by TFs in the same protein family as the predicted best match indicated in *Supplementary file 4*. TFs enriched in the Early active enhancer group show a decrease in expression over time while TFs enriched in the Late active enhancer group show an increase in expression over time. This correspondence between enriched TF expression and enhancer activity provides validation for our findings and indicates the timing of enhancer activity may be dictated by the expression and binding of these enriched TFs.

The top three enriched TF motifs for Early active enhancers were Ascl1, Meis2, and Atoh1 (*Figure 3A*). These TFs have established roles in cerebellar development, acting as markers of GABAergic or glutamatergic cell types and regulators of differentiation (*Ben-Arie et al., 1997*; *Kim et al., 2008*; *Wizeman et al., 2019*). Importantly, many of the motifs enriched in the Early group matched with TFs which have received little to no attention in the cerebellum, including Sox4, Lhx2, Rfx4, Pou3f1, and Pax3 (*Figure 3A*). These TFs have been previously associated, however, with the development of other brain areas (*Frantz et al., 1994*; *Porter et al., 1997*; *Su et al., 2016*; *Zhang et al., 2006*). In contrast, the TFs matching the motifs enriched in the Late active enhancers have a previously identified role in cerebellar development; but not necessarily in the same cellular processes (*Figure 3A*). For example, the top 3 enriched motifs matched with Neurod1, Nfia/b/x, and NF1, which have all been associated with granule cell differentiation (*Miyata et al., 1999*; *Sanchez-Ortiz et al., 2014*; *Wang et al., 2007*). However, two other TFs with enriched binding sites, Pax6 and Smad4 have been found to be critical for granule cell precursor proliferation, a process preceding differentiation

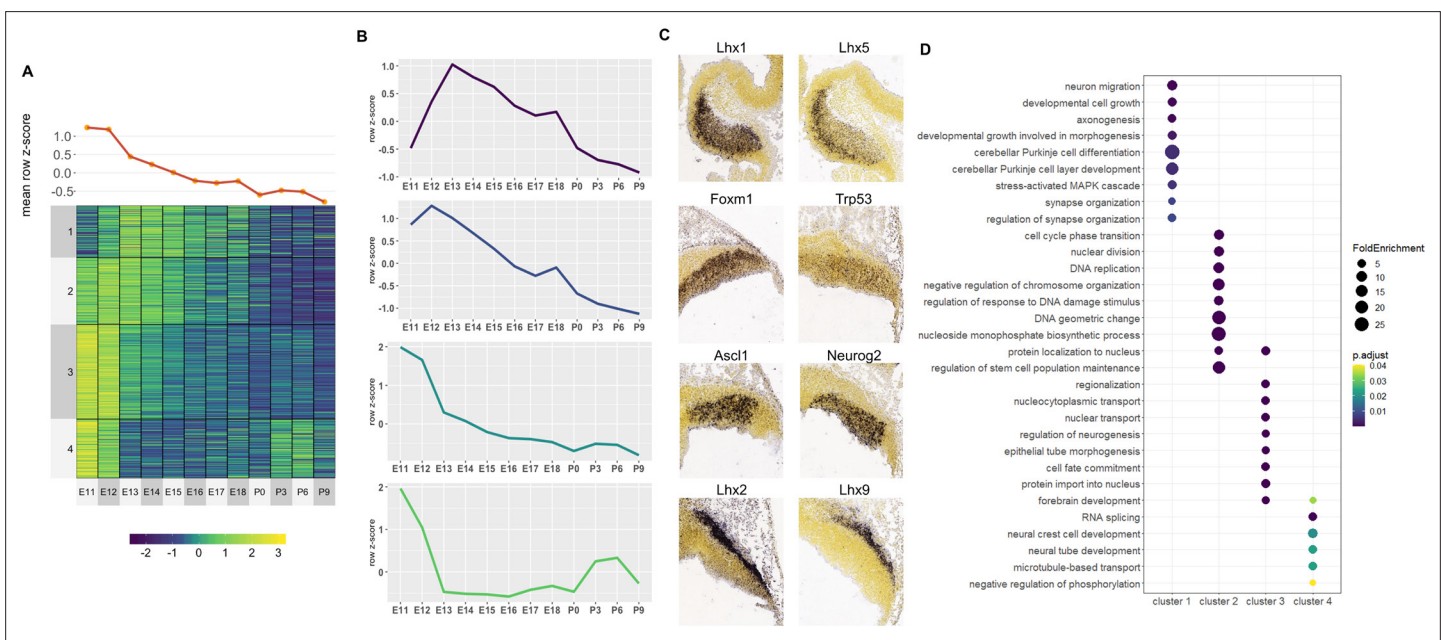

**Figure 4.** Correlated Early target genes are expressed in spatially distinct areas and have diverse roles in cerebellar development. (**A**) Line plot and heatmap showing mean z-score expression for Early target genes throughout the cerebellar time course. (**B**) Line graph representation of expression pattern throughout time for each cluster. (**C**) Known cerebellar genes in each cluster and in situ hybridization (ISH) images showing spatial expression at peak expression ages. ISH images were taken from the Developing Mouse Atlas at E13.5 for clusters 1 and 2, and E11.5 for clusters 3 and 4. (**D**) Gene Ontology (GO) enrichment analysis of target genes from each cluster, displaying the top enriched GO terms. Size of the dots indicates the observed vs expected fold enrichment of genes within that GO category (FoldEnrichment). Color scale indicates the adjusted p-value for each GO term.

The online version of this article includes the following figure supplement(s) for figure 4:

**Figure supplement 1.** Enrichment analysis of cerebellar genes in robust enhancer target gene list and Elbow anlaysis to determine the optimal k value for k-means clustering.

**Figure supplement 2.** Putative target genes of robust cerebellar enhancers containing Pax3 DNA binding motifs regulate neural progenitor function.

(*Swanson and Goldowitz, 2011*). These results suggest a dynamic role for the majority of our Early and Late active enhancers, driven by TFs involved in distinct stages of neuron development.

## Early active enhancers are enriched for Pax3 binding sites, a novel marker for GABAergic cells

The TF motif enrichment analysis of Early enhancers led to the discovery of several novel TFs in the context of embryonic cerebellar development; potentially involved in seminal aspects of development such as cellular specification or commitment. As a case study, we focused on Pax3, as other members of the Pax protein family have been shown to play key roles in the developing cerebellum (*Leto et al., 2009*; *Urbánek et al., 1997*; *Yeung et al., 2016*). Immunofluorescent staining was conducted to profile Pax3 expression in the embryonic and early postnatal mouse cerebellum. We observed robust expression in the ventricular zone (VZ); a neural progenitor region for GABAergic cells in the cerebellum (*Leto et al., 2006*; *Figure 3B*). A molecular marker of the VZ is Ptf1a, a GABAergic lineage-defining molecule in the cerebellum (*Hoshino et al., 2005*), and we examined if there was co-expression between Pax3 and Ptf1a. Colocalization between 50% of Pax3 positive cells in the VZ and Ptf1a (*Figure 3—figure supplement 1A*), (*Hoshino et al., 2005*) confirmed Pax3 expression within GABAergic neural progenitors. At E15, Pax3 + cells are seen in the region just dorsal to the VZ, which consist of post-proliferative cells such as Purkinje cells and interneurons (*Hoshino et al., 2005*; *Leto et al., 2006*; *Figure 4B*). We examined Pax3 co-labeling with markers for these cell types using Foxp2 (marker of Purkinje cells) and Pax2 (a general marker of GABAergic interneurons; *Fujita et al., 2008*; *Maricich and Herrup, 1999*). While colocalization between 91% of Pax3 + cells and Pax2 was found (*Figure 3B*, *Figure 3—figure supplement 1A*), no co-staining between Pax3 and Foxp2 was observed (*Figure 3—figure supplement 1B*). These results extend to P0 and P3, where Pax3 + cells are found in the inner granule cell layer (IGL) as well as the cerebellar parenchyma (*Figure 4B*, *Figure 3—figure supplement 1A*). To identify if Pax3 was expressed in interneuron precursors at these stages, co-labeling was conducted with Pax2. We found that 76% of Pax3 + cells co-localized with Pax2 and P0 and that 79% of Pax3 + cells co-localized with Pax2 at P3 (*Figure 4B*, *Figure 3—figure supplement 1A*). Pax3 did not colocalize with Calbindin, a Purkinje cell marker, at P0 (*Figure 3—figure supplement 1B*). At P9, Pax3 exhibited no observable expression (*Figure 3—figure supplement 1C*).

We then examined Pax3 expression throughout developmental time using a bulk tissue CAGE-seq dataset generated previously (*Ha et al., 2019*). Pax3 shows high expression during embryonic stages, peaking at E12, and decreasing steadily after E12 (*Figure 3—figure supplement 2A*). To further validate the cell types in which Pax3 are expressed, previously generated single-cell RNA-seq datasets quantified in the developing mouse and human cerebellum were examined (*Carter et al., 2018*; *Aldinger et al., 2021*). Pax3 showed similar expression profiles in the developing mouse and human cerebellum and is highly expressed in Lhx1/5+GABAergic progenitors and Pax2 +GABAergic interneurons supporting our immunofluorescent analysis (*Figure 3—figure supplement 2B-C*). Pax3 expression was also observed in other cell lineages, such as rhombic lip progenitors, UBC/Granule cell progenitors and astrocytes. We do not observe Pax3 expression through immunofluorescent staining in these cell types, which suggests a misalignment with single-cell data. This highlights the need to confirm the fidelity between histological and single-cell datasets. Taken together, our results indicate that Pax3 is a novel marker for GABAergic progenitors and interneuron precursors in the developing cerebellum.

## Co-expressed putative target genes are expressed in spatially distinct areas of the developing cerebellum

We next investigated the molecular processes regulated by robust cerebellar enhancers through predicting their downstream targets (*Osterwalder et al., 2018*; *Yao et al., 2015*). This was done by calculating the correlation between H3K27ac signal and gene expression at E12, P0, and P9 (*Ha et al., 2019*) for genes located within the same conserved topological associating domain (TAD) identified previously (*Dixon et al., 2012*) (See Materials and methods). Overall, at least one positively correlated target gene was identified for **5815/7023** (**70.61%**) cerebellar enhancers with an average Pearson correlation coefficient of **0.86** (*Supplementary file 5*). In total, we identified **2261** target genes. Using the Mouse Genome Informatics (MGI) database, we identified **98** target genes that when knocked out result in a cerebellar phenotype. To evaluate whether cerebellar genes were enriched in putative

target genes, we conducted a permutation analysis by generating 10,000 permutations of 2261 genes randomly selected from all genes expressed in the cerebellum and assessing the number of cerebellar genes in each permutation. We found that cerebellar genes were indeed enriched in target genes (p-value = 0.0405, *Figure 4—figure supplement 1A*) demonstrating the validity of our high-throughput approach.

An unbiased *k-means* clustering was then conducted for Early and Late target genes to delineate them into the various co-expression programs coordinating molecular events during development. For this analysis, the target gene expression time course was expanded to 12 different timepoints during cerebellar development, quantified previously by CAGE-seq (*Ha et al., 2019*). To determine the k-value, we conducted an Elbow analysis, identifying that 4 clusters were optimal for Early and Late active enhancers (*Figure 4—figure supplement 1B*).

For Early active enhancers, 4 Clusters of co-expressed target genes were identified (*Figure 4A*). Genes in these clusters had decreasing expression over time, similar to their corresponding enhancer activity. However, a distinct mean expression profile was observed for each Cluster (*Figure 4B*). Interestingly, genes with known function during cerebellar development showed distinct spatial expression patterns, observed using ISH data from the Developing Mouse Brain Atlas (*Thompson et al., 2014*; *Figure 4C*).

For example, in **Cluster 3**, cerebellar genes *Ascl1* and *Neurog2* are expressed exclusively in the ventricular zone at E11.5 while **Cluster 4** contains *Lhx9* and *Meis2* which are expressed in the Nuclear Transitory Zone (neurons destined for the cerebellar nuclei). A Gene Ontology (GO) enrichment analysis revealed that each cluster is enriched for molecular processes known to be regulated by cerebellar genes within the cluster (*Figure 4D*). For example, **Cluster 1** is enriched for **axonogenesis** (GO:0007409, p-value: 3.31E-4), **neuron migration** (GO:0001764, p-value: 3.3E-4) and **Purkinje layer development** (GO:0021691, p-value: 0.01) and also contains *Lhx1* and *Lhx5* which are expressed in migrating Purkinje cells in cerebellar parenchyma and has previously been associated with the regulation of Purkinje cell differentiation during embryonic cerebellar development (*Zhao et al., 2007*). Together, these findings support the notion that Early active enhancers regulate their targets in a spatially specific manner, regulating distinct processes in their respective cell types.

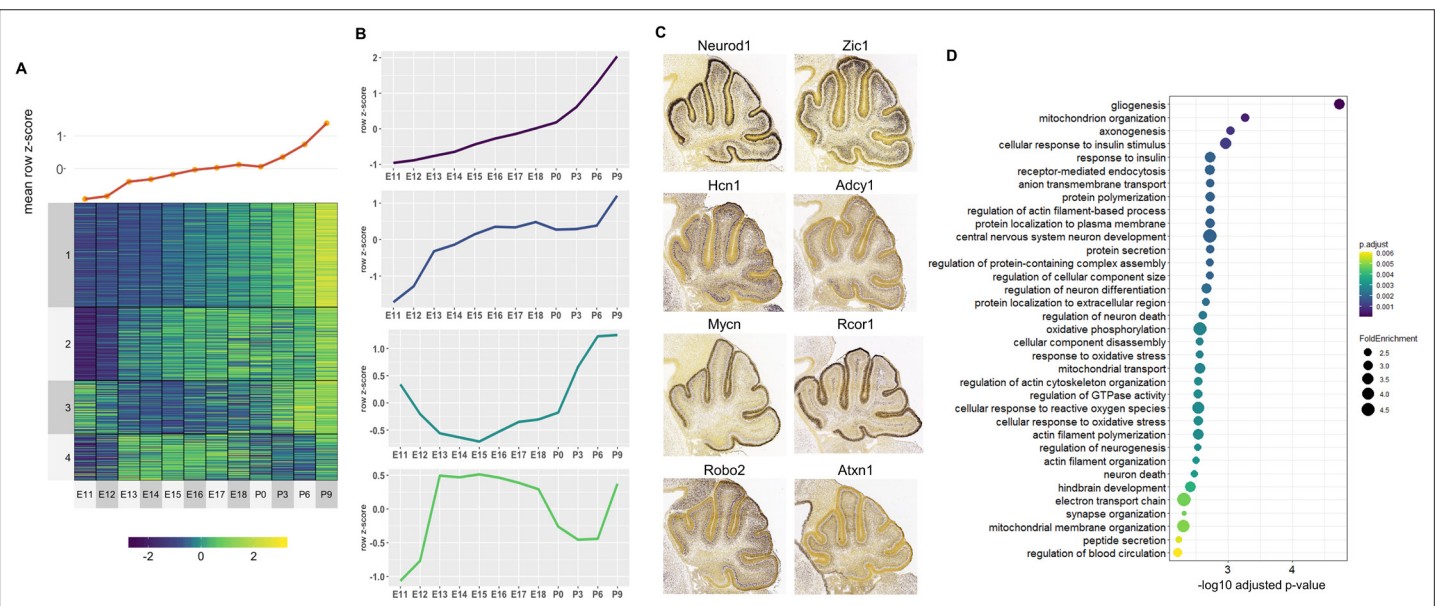

**Figure 5.** Correlated Late target genes are expressed in developing granule cells or Purkinje cells with common roles in cerebellar development. (**A**) Line plot and heatmap showing mean z-score expression throughout the cerebellar time course. (**B**) Line plot representation of expression pattern throughout time for each cluster. (**C**) Known cerebellar genes in each cluster and in situ hybridization showing spatial expression at peak expression ages. ISH images were taken from the Developing Mouse Atlas provided by the Allen Brain Atlas conducted at P4.5 for all clusters. (**D**) Gene Ontology (GO) enrichment analysis of all target genes of Late active enhancers, displaying the top enriched GO terms. Size of the dots indicates the observed vs expected fold enrichment of genes within that GO category (FoldEnrichment) and the x-axis represents the -log10 adjusted p-value for each GO term. Color scale indicates the adjusted p-value for each GO term.

We then used the results of our putative target gene analysis of Early active cerebellar enhancers to better understand the potential function of Pax3 during embryonic cerebellar development. We determined the putative target of genes of robust cerebellar enhancers containing Pax3 DNA binding motifs. We found that 1111 Early active robust cerebellar enhancers contained a Pax3 binding motif and that a putative target gene was identified for 923 Pax3-motif enhancers (*Supplementary file 6*). These target genes were enriched for GO terms pertaining to neural progenitor function and specification (*Figure 4—figure supplement 2*). Several putative Pax3 target genes have been previously associated with GABAergic neural progenitors in the cerebellar ventricular zone and the specification and differentiation of GABAergic cell types such as Ascl1 and Neurog2 (*Florio et al., 2012*; *Kim et al., 2008*). These genes also display a similar spatial expression pattern to Pax3 at E11, with high expression in the cerebellar ventricular zone (*Figure 4—figure supplement 2*).

For the Late active enhancers, 4 Clusters of co-expressed target genes were identified (*Figure 5A*). We observed relatively distinct expression patterns in each of the 4 Clusters with a gradual rise in mean expression over time (*Figure 5B*). Genes with known function during cerebellar development also show distinct spatial expression patterns, identified using the Developmental Mouse Atlas (*Figure 5C*). For example, **Clusters 1** and **3** contained known cerebellar genes critical for granule cell development, such as *Neurod1* and *Zic1*, while **Clusters 2** and **4** contained cerebellar genes important for in Purkinje cell development, such as *Atxn1* and *Hcn1* (*Figure 5C*; *Aruga and Millen, 2018*; *Ebner et al., 2013*; *Miyata et al., 1999*; *Rinaldi et al., 2013*).

A GO enrichment analysis was conducted for each Cluster; with no significantly enriched Cluster-specific GO terms. However, if all Late enhancer target genes were combined, several enriched GO terms emerged including ones involved in postnatal neuronal development, such as **neuron death** (GO:0070997, p-value: 0.003), **neurotransmitter transport** (GO:0006836, p-value: 0.006) and **regulation of synaptic vesicle exocytosis** (GO:2000300, p-value: 0.005) (*Figure 5D*). Overall, this analysis provides a working framework for the placement of hundreds of genes into the overall structure of embryonic or postnatal cerebellar development.

**Table 1.** A list of enhancer-regulated target genes from Late Cluster 1 found to be significantly differentially expressed in the conditional Atoh1 knockout mouse.

The second and third column contain the observed P-value and fold change from the differential expression analysis, respectively. The fourth and fifth columns indicate whether the gene has previously been implicated in cerebellar development and the corresponding reference PubMed ID.

| Gene | p-value (Atoh1-null) | Fold Change (Atoh1-null) | Cerebellar Development | Reference (PMID) |
|---|---|---|---|---|
| Neurod1 | 9.196E-229 | 0.2 | X | 19609565 |
| Nfix | 1.1991E-43 | 0.53 | X | 21800304 |
| Zic1 | 1.3082E-37 | 0.35 | X | 21307096 |
| Barhl1 | 2.2505E-35 | 0.22 | X | 9412507 |
| Zic2 | 2.0968E-20 | 0.34 | X | 11756505 |
| Insm1 | 5.4814E-20 | 0.25 | X | 18231642 |
| Tcf4 | 9.9139E-20 | 0.69 | X | 30830316 |
| Nfia | 1.8675E-16 | 0.6 | X | 17553984 |
| **Bhlhe22** | **4.7662E-10** | **0.53** | | |
| **Purb** | **2.5878E-09** | **0.53** | | |
| Neurod2 | 4.424E-09 | 0.37 | X | 11356028 |
| **Klf13** | **1.7938E-06** | **0.72** | | |
| Zfp521 | 3.5899E-06 | 0.8 | X | 24676388 |
| **Sox18** | **3.7168E-05** | **0.71** | | |
| Nfib | 0.00021009 | 0.63 | X | 17553984 |

## *Bhlhe22* is a novel regulator of granule cell development

To demonstrate the utility of our results, we sought to elucidate target genes not previously identified in cerebellar development. We focused on Late Cluster 1, which contained target genes expressed in granule cells. We hypothesized that genes within this cluster regulated postnatal granule cell differentiation. To identify genes in this cluster regulating granule cell development, we filtered these genes relative to their interaction with Atoh1, the lineage defining molecule for granule cells and other glutamatergic neurons in the developing cerebellum (*Ben-Arie et al., 1997*). The genes were filtered using the following criteria: (1) Atoh1 is bound to the predicted enhancer during postnatal development (*Klisch et al., 2011*) and (2) the genes are differentially expressed in the Atoh1-null mouse (*Klisch et al., 2011*). The resulting list of genes were sorted based on differential expression in the Atoh1-null mouse and filtered for TFs. These criteria filtered gene candidates for validation from Late Cluster 1 from 254 genes to 26 genes. Among the top 15 genes in the filtered list, we identified **4** novel genes and **11** genes that have previously been implicated in postnatal granule cell development (*Table 1*).

The known genes provided validation for our approach. The novel genes included *Bhlhe22* (also known as *Bhlhb5*), *Purb*, *Klf13*, and *Sox18*. We focused on *Bhlhe22* as it has previously been implicated in the differentiation of neurons in the cortex (*Joshi et al., 2008*). An enhancer ~2 kb upstream of the *Bhlhe22* transcriptional start site was identified and is bound by Atoh1 during postnatal development (*Figure 6—figure supplement 1A*). This enhancer displayed H3K27ac activity highly correlated

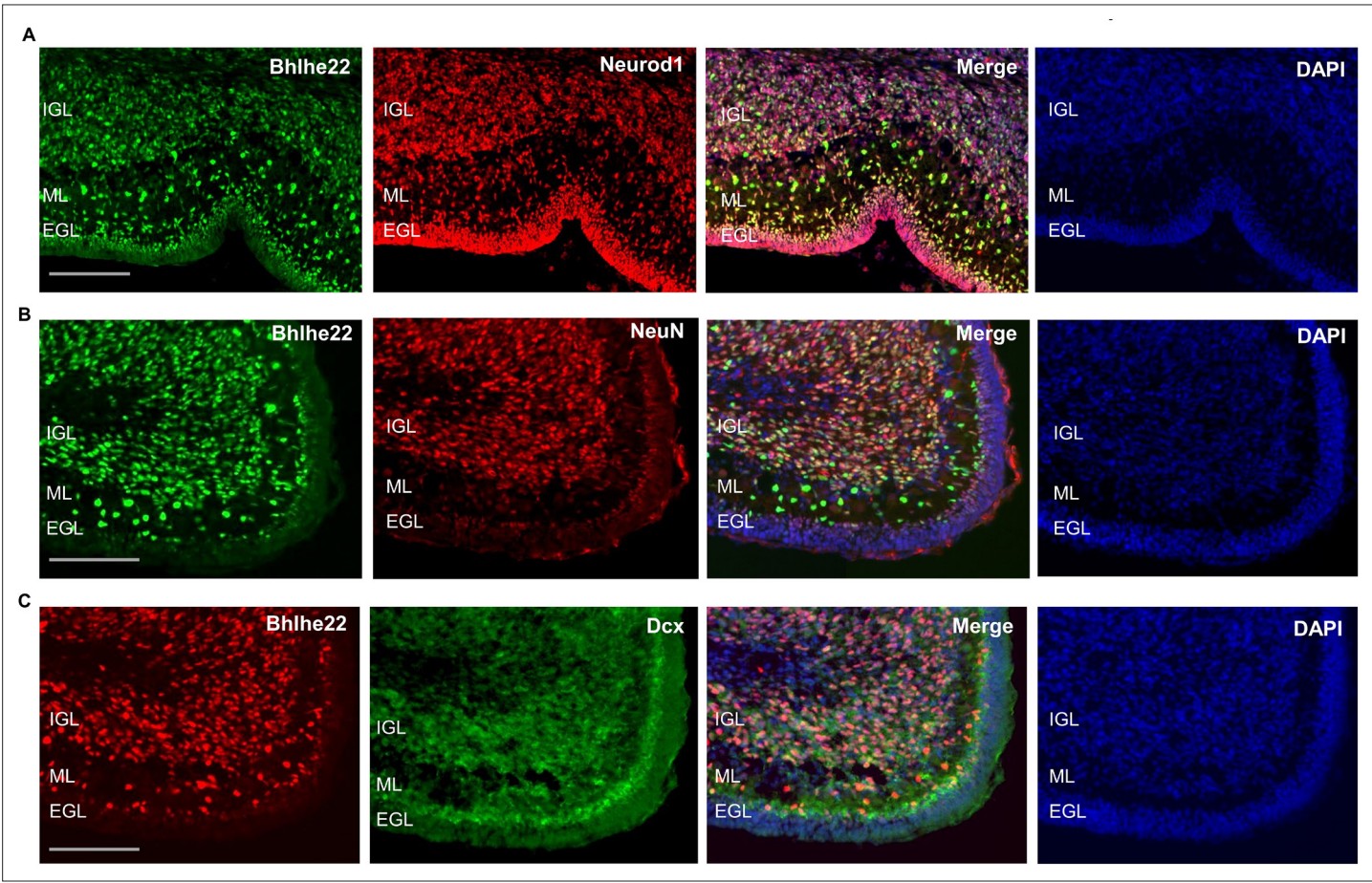

**Figure 6.** Bhlhe22 is expressed in differentiating granule cells in postnatal cerebellar development. (**A**) Bhlhe22 (green) and Neurod1 (red) immunofluorescent co-staining at P9.5 of taken from a posterior lobe IX. (**B**) Bhlhe22 (green) and NeuN (red) immunofluorescence co-staining at P6 taken from posterior lobe IX. (**C**) Bhlhe22 (red) and Dcx (green) immunofluorescent co-staining at P6 showing the posterior lobe IX; Labels: EGL = external granular layer IGL = internal granular layer, ML = molecular layer, Scalebars = 100 µm.

The online version of this article includes the following figure supplement(s) for figure 6:

**Figure supplement 1.** Transcription factor Bhlhe22 is expressed in the granule cell layer during cerebellar development.

**Figure supplement 2.** Bhhe22 expression in the developing mouse and human cerebellum quantified by scRNA-seq.

(Pearson correlation coefficient = 0.98) to *Bhlhe22* expression, which consistently rises throughout cerebellar development and peaks at P9.5 (*Figure 6—figure supplement 1B*).

To attain a cellular resolution of the expression pattern for Bhlhe22 during postnatal cerebellar development cerebellar development, immunofluorescent staining was conducted. Bhlhe22 expression was observed in cells within the inner external granule layer (EGL), molecular layer (ML) and in the internal granule layer (IGL) of the postnatal cerebellum (*Figure 6A*). Cells in the ML are Bhlhe22 +may be of two phenotypes: (1) small cell-bodied radially oriented migrating granule cells and (2) larger cell-bodied rounded interneurons.

To identify whether Bhlhe22 is expressed in differentiating granule cells, co-staining experiments were performed with Neurod1 and NeuN which mark differentiating and more mature granule cells, respectively (*Miyata et al., 1999*; *Weyer and Schilling, 2003*). At P6.5, colocalization between 98% of Bhlhe22 + cells and Neurod1 was observed, indicating expression in differentiating and migrating granule cells (*Figure 6A*, *Figure 6—figure supplement 1C*). Co-staining between 82% of Bhlhe22 + cells and NeuN expression was also observed, indicating expression in maturing granule cells found within the IGL (*Figure 6B*, *Figure 6—figure supplement 1C*). To confirm whether the Bhlhe22-positive cells within the molecular layer were migrating granule cells, we performed a double labelling experiment for a neuronal migration marker Doublecortin (*Takács et al., 2008*). Colocalization of Doublecortin in 95% of Bhlhe22 + cells was observed within the inner EGL and the molecular layer, confirming Bhlhe22 expression in migrating granule cells (*Figure 6C*, *Figure 6—figure supplement 1C*). To assess Bhlhe22 expression at a single-cell resolution, we examined Bhlhe22 expression in

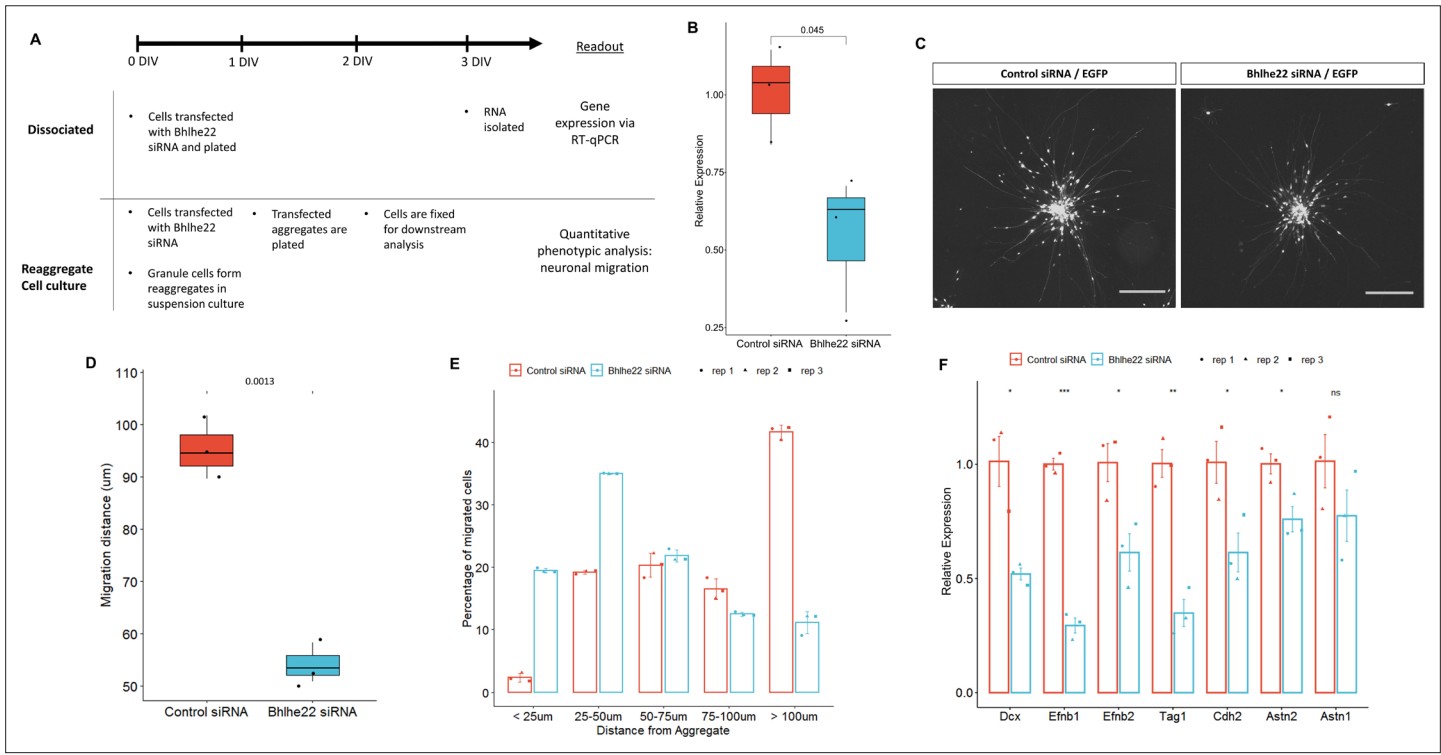

**Figure 7.** Knockdown of Bhlhe22 reduces migration of cultured cerebellar granule cells. (**A**) Workflow for dissociated and reaggregate postnatal granule cell cultures. (**B**) RT-qPCR analysis of Bhlhe22 gene expression in dissociated postnatal granule cell cultures after treatment with Bhlhe22 siRNA. Gene expression was normalized relative to the expression of the co-transfected EGFP protein to account for transfection variability between cultures. Data are presented as mean ± SD (n=3). (**C**) Image of cultured cerebellar granule cell reaggregates treated with control and Bhlhe22 siRNA. Shown are EGFP-positive cells indicating successful transfection. Scalebars = 100 μm. (**D**) Box plot displaying mean distance of granule cell migration from the aggregate. Value above indicates a statistical difference between control cultures and those treated with Bhlhe22 siRNA (p-value = 0.0013). (**E**) Bar plot showing the percentage of cells migrated at different distances from the aggregate for control and Bhlhe22 siRNA-treated cerebellar granule cell cultures. (**F**) RT-qPCR analysis of gene expression of cell adhesion molecules in dissociated postnatal granule cell cultures after treatment with Bhlhe22 siRNA. Gene expression was normalized relative to the expression of the co-transfected EGFP protein to account for transfection variability between cultures. Data are presented as mean ± SD (n=3). Symbols: *: p ≤ 0.05, **: p ≤ 0.01, ***: p ≤ 0.001, which indicate statistical differences observed between Bhlhe22 siRNA-treated samples and controls. All error bars represent the standard error of the mean,.

single-cell RNA-seq datasets previously generated in the developing mouse and human cerebellum (*Carter et al., 2018*; *Aldinger et al., 2021*). In developing mouse and human cerebellum, Bhlhe22 is highly expressed in granule cells and granule cell progenitors, supporting the results of our immuno-fluorescent analysis (*Figure 6—figure supplement 2*). Bhlhe22 is also expressed in other cell types in the glutamatergic lineage, such as unipolar brush cells and excitatory cerebellar nuclei as well as high expression in inhibitory GABAergic interneurons neurons indicating importance for both neuronal lineages in the cerebellum.

We then investigated the role that Bhlhe22 plays in postnatal granule cell development using a well-established in vitro system (*Lee et al., 2009*). Three sets of experiments were performed using P6.5 isolated granule cells transfected with siRNA targeting Bhlhe22 transcripts (*Figure 7A*). First, to determine if Bhlhe22 expression was diminished, changes in gene expression were assessed after 3 days in vitro (DIV) using reverse transcriptase quantitative PCR (RT-qPCR). A 50% reduction of Bhlhe22 expression, on average, was found in treated cultures compared to controls (*Figure 7B*).

Second, phenotypes of the transfected cells were examined: neuritic outgrowths from the aggregate and the migration of granule cells from the aggregates, within the first 24 hr of plating (*Gärtner et al., 2006*). Neuritic outgrowth was unaffected. In contrast, a marked reduction in migration was found (*Figure 7C*). Bhlhe22 siRNA transfected cells travelled on average 54.2 µm from the edge of the aggregate, a 50% reduction compared to control samples (*Figure 7D*). Furthermore, examining the distribution of migrated cells from the edge of the aggregate identified a higher percentage of Bhlhe22 siRNA transfected granule cells migrating less than 50 µm, while the majority of the cells in control samples migrated 100 µm and beyond (*Figure 7E*).

Third, changes in the expression of cell adhesion molecules that are known to be involved in granule cell development were assessed (*Consalez et al., 2020*; *Wang et al., 2007*). A significant reduction of *Efnb1, Efnb2, Tag1, Cdh2,* and *Astn2* was observed in Bhlhe22 knockdown granule cell cultures compared to controls (*Figure 7F*). In addition to these genes, we also found a significant reduction in Doublecortin (*Dcx*) expression. Taken together, these in vitro knockdown experiments reveal a novel function for Bhlhe22, a gene that was identified by our temporal enhancer-target gene analysis and was predicted to have a critical role in postnatal granule cell development.

## Active cerebellar enhancers are enriched for common and de novo genetic variants associated with autism spectrum disorder

Given the emerging importance of the cerebellum in the etiology of autism spectrum disorder (ASD) (*Limperopoulos et al., 2014*; *Stoodley and Limperopoulos, 2016*) and to demonstrate the utility of our dataset in potentially functionally annotating variation associated with neurodevelopmental disorders, we tested whether ASD-associated variants and de novo mutations are enriched in cerebellar enhancers. Analysis of genome wide association studies (GWAS) have revealed that the majority of variants associated with neurodevelopmental diseases are found within non-coding regulatory sequences, particularly enhancers (*Visel et al., 2009*). The software tool GREGOR (Genomic Regulatory Elements and Gwas Overlap algoRithm) was used to evaluate the enrichment of common genetic variants associated with ASD in robust cerebellar enhancers (*Schmidt et al., 2015*). Robust cerebellar enhancers were converted from mouse (mm9) to human (hg38) genomic coordinates (6630/7024, 94.4% converted). The majority (89.6%) of robust cerebellar enhancers with a putative target gene are located at orthologous regions distal to the same gene promoters indicating conservation between mouse and humans and bringing credence to our analysis (*Supplementary file 7*). ASD-associated SNPs were retrieved from the GWAS Catalog (*Buniello et al., 2019*) and a stringent filter was applied to identify SNPs associated with the ASD (see Materials and methods). We examined 174 ASD-associated SNPs with a maximum p-value of 9E-06 (*Buniello et al., 2019*). ASD-associated SNPs were enriched in robust cerebellar enhancers (p-value = 2.34E-03) and in H3K27ac peaks at E12, P0, and P9 (p-values of 1.29E-03, 1.05E-02 and 1.42E-04, respectively) (*Figure 8A*). As a negative control, we conducted the same analysis with SNPs associated with chronic kidney disease and found no enrichment in cerebellar enhancers. For the 13 cerebellar enhancers containing ASD-associated SNPs, we identified 12 predicted target genes (*Supplementary file 8*). Among these, three (*PAX6*, *TCF4*, and *ZMIZ1*) are ASD risk genes according to the Simons Foundation Autism Research Initiative (SFARI) gene database (*Banerjee-Basu and Packer, 2010*).

**A**

| Dataset | ASD Reported Number of SNPs | ASD Expected Number of SNPs | ASD P-value | Kidney Disease Reported Number of SNPs | Kidney Disease Expected Number of SNPs | Kidney Disease P-value |
|---|---|---|---|---|---|---|
| Cerebellar enhancers | 13 | 5.31 | 2.34E-03 | 15 | 11.89 | 0.211 |
| E12 H3K27ac | 27 | 15.40 | 1.29E-03 | 33 | 34 | 0.625 |
| P0 H3K27ac | 5 | 1.37 | 1.05E-02 | 1 | 2.99 | 0.948 |
| P9 H3K27ac | 19 | 7.94 | 1.42E-04 | 16 | 26.97 | 0.639 |

**B**

| Dataset | Number of DNMs | | | | Odds ratio (confidence interval) | P-value |
|---|---|---|---|---|---|---|
| | Enhancer affected | Enhancer unaffected | Non-enhancer affected | Non-enhancer unaffected | | |
| Cerebellar enhancers | 1605 | 609 | 389636 | 161992 | 1.10 (1.00, 1.21) | 0.055 |
| E12 H3K27ac peaks | 4321 | 1726 | 386920 | 160875 | 1.04 (0.98, 1.10) | 0.16 |
| P0 H3K27ac peaks | 337 | 132 | 390904 | 162469 | 1.06 (0.87, 1.31) | 0.58 |
| P9 H3K27ac peaks | 2041 | 796 | 389200 | 161805 | 1.07 (0.98, 1.16) | 0.13 |
| Combined Coordinates | 7712 | 3016 | 383529 | 159585 | 1.06 (1.02, 1.11) | 0.0043 |

**C**

| Enhancer Coordindates (hg38) | Predicted Target Gene | CNV Coordinates (hg38) | CNV Type | CNV Size (bp) |
|---|---|---|---|---|
| chr10:13928845-13929693 | FAM107B | chr10:13912983-13942132 | Deletion | 29,150 |
| chr11:63909702-63910591 | STIP1 | chr11:63907636-63910623 | Deletion | 2,988 |
| chr12:96838867-96841675 | NEDD1 | chr12:96825001-96842000 | Duplication | 17,000 |
| chr14:76714160-76716619 | VASH1 | chr14:76711247-76714368 | Deletion | 3,122 |
| chr14:103861041-103862026 | CDC42BPB | chr14:103855620-103866732 | Deletion | 11,113 |
| chr15:98960270-98962257 | - | chr15:98937001-99031000 | Duplication | 94,000 |
| chr19:47156912-47157972 | MEIS3 | chr19:47133001-47184000 | Duplication | 51,000 |

**D**

Cdc42bpb

**Figure 8.** Cerebellar enhancers are enriched for GWAS SNPs and DNMs associated with ASD. (**A**) Enrichment analysis of ASD-associated and chronic kidney disease associated (negative control) GWAS variants in cerebellar enhancers and H3K27ac peaks called from E12, P0, and P9 samples. (**B**) Enrichment of de novo single nucleotide variants and indels in ASD-affected individuals compared with their unaffected siblings. Counts are not equal to the sum of the four enhancer types because some enhancers are categorized as more than one type. (**C**) Gene targets for enhancers overlapped by de novo CNVs in the SSC cohort. (**D**) Left: Line graph representing Cdc424bpb normalized expression in the developing mouse cerebellum from E11 to P9. TPM = transcripts per million. **Right:** In situ hybridization showing Cdc42bpb expression in the lateral (left) and medial (right) adult mouse cerebellum (Developing Mouse Brain Atlas). Note expression is found in granule cells, particularly those of the lateral cerebellum.

The online version of this article includes the following figure supplement(s) for figure 8:

**Figure supplement 1.** Integrative Genomics Viewer visualization of a de novo 11 kb deletion overlapping an enhancer predicted to target the *CDC42BPB* gene.

**Figure supplement 2.** Sequence alignment between mouse and humans for CDC42BPB and an enhancer element predicted to regulated CDC42BPB expression.

**Figure supplement 3.** Cdc42bpb expression in the developing mouse and human cerebellum quantified by scRNA-seq.

De novo mutations (DNMs) (variants present in the genome of a child but not his or her parents) have been found to play a significant role in the etiology of ASD, including those found in non-coding regions of the genome (*Grove et al., 2019*; *Yuen et al., 2016*). We hypothesized that DNMs within cerebellar enhancers would be more prevalent in ASD-affected individuals compared with their unaffected siblings. We used whole-genome sequencing data from 2603 ASD-affected individuals and 164 unaffected siblings from the MSSNG cohort (*C Yuen et al., 2017*), as well as 2340 ASD-affected individuals and 1898 unaffected siblings from the Simons Simplex Collection (SSC) (*Fischbach and Lord, 2010*) to analyze the prevalence of DNMs in ASD-affected individuals compared with their unaffected siblings.

We found that DNMs (specifically de novo single nucleotide variants and indels) in cerebellar enhancers and H3K27ac peaks from E12, P0, and P9 were enriched in ASD-affected individuals, with odds ratios ranging from 1.04 to 1.10 (*Figure 8B*). While these differences were not statistically significant for cerebellar enhancers and peak coordinates individually, statistical significance was achieved when combined (odds ratio = 1.06; *P*-value = 0.0043). We also identified de novo CNVs overlapping cerebellar enhancers. Since the number of such CNVs was too small to perform statistical enrichment tests, we selected a subset of seven of these CNVs (four deletions and three duplications) for further characterization to identify candidates for association with ASD (*Figure 8C*). The most promising candidate was an ~11 kb deletion overlapping an enhancer predicted to regulate *CDC42BPB* (803 kb upstream of the TSS), which has previously been implicated in neurodevelopmental phenotypes

(*Chilton et al., 2020*). By visual validation in Integrative Genomics Viewer (*Robinson et al., 2011*), we verified that this deletion was truly de novo (*Figure 8—figure supplement 1*). Within these coordinates, there is a proportion of sequence alignment between mouse and human genomes located within a robust cerebellar enhancer on chromosome 12, and this enhancer is located in cis a similar distance from the Cdc42bpb TSS as in humans (833 kb) indicating that enhancer activity occurs at orthologous regions (*Figure 8—figure supplement 2A*). Cdc42bpb is expressed steadily throughout mouse cerebellar development and is expressed in the granule cell layer of the adult mouse cerebellum (*Figure 8D*). Expression in granule cells is found in the lateral aspects of the adult cerebellum but not the medial adult cerebellum. In the context of cerebellar development, scRNA-seq data from mice and humans showed relatively consistent expression levels across most cell types, with highest expression in glutamatergic cerebellar nuclei and developing Purkinje cells (*Figure 8—figure supplement 3A*).

## Discussion

In our study on the cerebellum, we performed a novel assessment of enhancer activity through genome-wide profiling of H3K4me1 and H3K27ac at three time points during embryonic and early postnatal stages. These datasets were utilized to define functional enhancer elements with temporally specific activity during these developmental ages. The biological processes under enhancer regulation were described through motif enrichment analysis and target gene prediction, identifying temporally and spatially specific regulatory functions. As a result, our study provides a novel dataset for the developmental biology and neuroscience communities, especially those interested in functionally annotating enhancers in the context of brain development. We have created an online resource that can be used to access, curate and export our dataset. In addition to genomic coordinates of our cerebellar enhancers, this resource provides the activity patterns (Early or Late) for each enhancer throughout developmental time, as well as putative target genes. Predicted cell-types in which a robust cerebellar enhancer may be active as well as activity patterns in other brain regions are also included in our resource based upon comparisons with previously generated datasets (ENCODE and single-cell chromatin accessibility). As demonstrated in our analysis of Pax3 and Bhlhe22 in the context of the cerebellum, this resource can facilitate the discovery of novel genetic regulators of cerebellar development.

### Cerebellar enhancers regulate gene expression important for distinct stages of neuronal development

Identification of enriched TF binding sites and putative target genes indicated that cerebellar enhancers likely play a regulatory role in various phases of neuronal development. In agreement with our results, previous examinations of active non-coding regulatory sequences revealed that neural progenitor cells and mature neurons exhibit distinct signatures of enhancer-associated histone profiles, DNA methylation, chromatin conformation and enhancer-promoter interactions (*Bonev et al., 2017*; *Torre-Ubieta et al., 2018*; *Lister et al., 2013*; *Whyte et al., 2012*). *Bonev et al., 2017* examined changes in enhancer-promoter interactions between transgenic cell lines that were FACS sorted for embryonic stem cells, neural progenitors and mature neurons and identified that changes in enhancer-promoter contacts are cell-state specific and correlate with changes in gene expression (*Bonev et al., 2017*). A global shift in regulatory sequence usage was observed between neuro-progenitors and mature neurons, indicated by dynamic changes in enhancer-promoter interactions. These changes were also reflected at the level of TF binding, as interactions at Pax6-bound sites, a TF marking neural progenitors, were stronger in neural progenitors than in neurons, while NeuroD2-bound sites, a TF marking mature neurons, were stronger in neurons than NPCs (*Bonev et al., 2017*). This shift in enhancer usage throughout cortical development is also reflected in DNA methylation profiles, where fetal enhancers are hypermethylated and decommissioned in the adult brain, while enhancers regulating adult gene expression were hypomethylated (*Lister et al., 2013*). Hypermethylation was accompanied by a decrease in H3K4me1, H3K27ac and DNase hypersensitivity while the increase was observed after hypomethylation (*Lister et al., 2013*). Our study supports the importance of temporally-specific activity during different stages of neuron development in vivo and details the processes driven by enhancer-regulated expression during embryonic and early postnatal brain development.

Expression analysis of two genes novel to cerebellar development, Pax3 and Bhlhe22, supported the notion that enhancer profiles are specific to developmental stage. TF enrichment analysis identified Pax3 preferentially enriched in Early active enhancers and robust expression of Pax3 was localized to GABAergic interneuron progenitor cells. Pax3 has previously been associated with neural tube and neural crest development (*Epstein et al., 1991*; *Olaopa et al., 2011*); however, the study of it's function in cerebellar development is limited. Pax3 has been shown to be upregulated by BDNF and NGF in cerebellar in vitro cultures (*Kioussi and Gruss, 1994*). In the context of human disorders and disease with cerebellar phenotypes, it has been found to be downregulated Group 3 medulloblastomas (*Zagozewski et al., 2020*) and is located downstream of a deletion on chromosome 2 in Dandy-Walker Syndrome patients (*Jalali et al., 2008*). Analysis of predicted gene targets of Late active enhancers identified Bhlhe22 as a novel gene expressed in postnatal differentiated granule cells, and in vitro knockdown experiments in primary granule cells indicated Bhlhe22 regulates granule cell migration potentially through regulation of cell adhesion molecule expression. These results are supported by findings in the developing cortex, where Bhlhe22 has been shown to regulate post-mitotic acquisition of area identity in layers II-V of the somatosensory and caudal motor cortices (*Joshi et al., 2008*). The contrasting expression profiles of Pax3 and Bhlhe22 highlight the wide-ranging developmental impact of enhancer-mediated gene expression regulation.

## Co-expressed gene targets of cerebellar enhancers display cell-type-specific expression patterns

In addition to being temporally-specific, recent evidence indicates that enhancer activity is cell type specific in the brain (*Blankvoort et al., 2018*). In the context of cerebellar development, a recent study examining open chromatin regions genome-wide using snATAC-seq identified a catalog of cerebellar cis-regulatory elements (CREs) and has revealed that the cell types of the cerebellum have unique chromatin signatures throughout embryonic and postnatal development (*Sarropoulos et al., 2021*). We found that the majority (~90%) of the robust cerebellar enhancers identified in our study overlap these CREs. Additionally, open chromatin signals at these sequences closely resemble the changes in chromatin marks for Early and Late cerebellar enhancers in the major neuronal cell types in the cerebellum. This overlap further validates our findings and provides additional evidence for enhancer activity at these genomic locations. The comparison of our results with CREs identified in single cells allowed the prediction of the cell types in which our robust cerebellar enhancers may be active. These predictions will help guide future studies looking to validate enhancer activity and regulatory function.

Spatial specificity of enhancer activity is also highlighted in the identification of cerebellar enhancer target gene clusters for Early and Late active enhancers with cell specific patterns of expression (*Figures 4 and 5*). For example, distinct boundaries can be seen in gene expression from Early Clusters 3 and 4 at E11.5 between cells in the subpial stream (Cluster 4) and neuroepithelium (Cluster 3) where neural precursors of two separate lineages, the glutamatergic cerebellar nuclei and GABAergic cerebellar nuclei and Purkinje cells, are found, respectively. These sharp borders are reminiscent of the small domains of distinct enhancer activity identified in neural progenitors in the telencephalon, which were found to fate-map to specific prefrontal cortex subdivisions (*Pattabiraman et al., 2014*). We see a similar pattern in the more developed postnatal cerebellum, observing a spatial distinction between Late Clusters 1/3 and 2/4 delineating expression in granule cells and Purkinje cells, respectively. This cell-type-specific enhancer usage is demonstrated in the adult brain. *Blankvoort et al., 2018* used ChIP-seq analysis of microdissected subregions of the adult mouse cortex to reveal unique enhancer profiles pertaining to each region. Additionally, *Nott et al., 2019* identified enhancer-promoter interactome maps specific to the major cell types in the cortex, which included neurons, microglia, oligodendrocyte, and astrocytes. Enriched GO terms for each cerebellar target gene clusters were cell-type and temporally specific, highlighting enhancer specificity. Functionally annotating their respective clusters provides a working hypothesis for hundreds of genes, which can be used as a jumping point for future in-depth studies in the cerebellum. Collectively, these findings support the notion that the cell types in the cerebellum have specific enhancer signatures which are reflected by the expression and functions of their target genes.

The results from our analyses revealed that the majority of the identified cerebellar enhancers are predicted to be active in the most abundant cell types at each developmental stage. For example, most Early active enhancers were predicted to be active in progenitors while Late active enhancers

were predicted to be active in the developing granule cells and Purkinje cells. We attribute this apparent bias to our bulk tissue approach, which inherently identifies active enhancers based on signal abundance (ie. peak calling) making it more likely to capture signals specific to more abundant cell types compared to other less abundant and rare cell types. In order to assess histone marks in less abundant cell types, a more granular approach should be implemented such as isolation of certain cell types or single-cell analysis. Recent technological developments have allowed the examination of histone modifications genome-wide in single cells, which has seen preliminary success in mouse brain tissue (*Zhu et al., 2021*, *Bartosovic et al., 2021*, *Rang et al., 2022*).

## Cerebellar enhancers display activity in other developing brain regions, especially during embryonic development

The comparison between active cerebellar enhancers with histone profiles generated in other developing brain regions revealed that the majority of Early enhancers were active in multiple brain regions. Previous genome-wide and locus-specific functional studies have shown evidence of enhancers with activity in multiple developmental contexts (*Singh and Yi, 2021*; *Preger-Ben Noon et al., 2018*; *Lonfat et al., 2014*; *Andersson et al., 2014*; *Nord et al., 2013*; *Hiller et al., 2012*). For example, using H3K27ac as a predictor of enhancer activity across multiple tissues, *Nord et al., 2013* identified that 52% of predicted enhancers are active in more than one organ, of which 31% are active in two organs and 21% are active in three organs. Additionally, these studies have also revealed that pleiotropic enhancers may serve as binding sites for TFs that may be reused in different contexts. In *Drosophila*, *Preger-Ben Noon et al., 2018* found that enhancer elements regulating the *shavenbaby* gene drive expression in multiple tissues and developmental stages. In one of these enhancers, the same TF binding site is used during both embryonic and pupal expression, while another enhancer utilizes different binding sites. In the context of brain development, a study that compared the binding of p300 at enhancers identified pleiotropic activity in both the embryonic mouse forebrain and midbrain at E11.5 (*Hiller et al., 2012*). Overall, a more detailed examination of enhancer activity across brain sub-regions and validation of the bound TFs and downstream target genes of enhancers with pleiotropic activity in the brain in figure studies will provide insight into the functional role these enhancers serve. Single-cell analysis of enhancer activity, as mentioned above, may provide further insight on how often enhancers function across cell types in the developing brain and the timing in which they are active.

## GWAS SNPs and DNMs associated with ASD are enriched in cerebellar enhancers

Having established and characterized enhancer sequences in the cerebellum, we sought to elucidate the potential involvement of these regions in the etiology of neurological disorders and demonstrate how our dataset may be utilized to functionally annotate variation associated with neurodevelopmental disorders; imaging and quantitative data show consistent cerebellar abnormalities, particularly in cases of individuals with autism (*Limperopoulos et al., 2014*; *Stoodley and Limperopoulos, 2016*). Our results indicate that cerebellar enhancer sequences are significantly enriched for GWAS variants and DNMs associated with ASD, suggesting an important role for enhancers in contributing to the condition. PAX6 was among 12 target genes of cerebellar enhancers containing ASD-associated variants and is classified in the SFARI database as a ASD risk gene. The deletion of Pax6 in the murine cerebellum results in aberrant development of the glutamatergic cells in the cerebellum: the cerebellar nuclei, unipolar brush cells, and granule cells (*Yeung et al., 2016*). Behavioral analysis of Pax6 animal models has also indicated a possible link between this gene and autistic-like behavior (*Umeda et al., 2010*). Additionally, Pax6 has been linked with WAGR (Wilm's tumor, Aniridia, Genitourinary malformations, and mental Retardation syndrome) which is co-morbid for ASD. Our analysis invites future investigation these target genes and how perturbation of their expression may lead to ASD phenotypes.

Of the target genes of the enhancers that overlapped de novo CNVs in the SSC cohort, none have been previously associated specifically with cerebellar development. Interestingly, one of these target genes, *CDC42BPB,* has recently been associated with neurodevelopmental disorders including ASD (*Chilton et al., 2020*). This gene is a serine/threonine protein kinase and codes for MRCKβ (myotonic dystrophy-related Cdc42-binding kinase beta), a regulator of cell cytoskeletal reorganization and cell

migration (*Pichaud et al., 2019*). Of note, the CNV associated with this gene deletes the entire enhancer. CDC42BPB shows expression in the granule cell layer of the lateral adult cerebellum, which has been associated with cognitive functions (*Koziol et al., 2014*).

Together, our dataset provides an atlas of active enhancers during embryonic and postnatal cerebellar development. Not only have we demonstrated the utility of our dataset through the discovery of TFs with novel functions in the developing cerebellum, we also provide an invaluable resource for future studies. The online Developmental Cerebellum Enhancer Atlas can be used by the scientific community as a confirmatory tool for ongoing studies and to identify novel candidate genes involved in cerebellar development for future studies.

## Materials and methods
### Mouse strains and husbandry
C57BL/6 J mice were originally purchased from JAX laboratory and maintained and bred in our pathogen-free animal facility with 12/12 hour light/dark cycle and a controlled environment. Embryonic ages utilized in these experiments were confirmed based upon the appearance of a vaginal plug. The morning that a vaginal plug was detected was designated as E0.5. Pregnant females were cervically dislocated and embryos were harvested from the uterus. Postnatal ages were determined based upon the date of birth with the morning of the observation of newborn pups considered as P0.5. All studies were conducted according to the protocols approved by the Institutional Animal Care and Use Committee and the Canadian Council on Animal Care at the University of British Columbia.

### Tissue preparation for chromatin immunoprecipitation
C57BL/6 J mice (male and female) at E12.5, P0.5 and P9.5 (referred to as E12, P0, and P9) were decapitated for dissection of cerebella. Cerebella were dissected and collected in ice cold Dulbecco's PBS (DPBS) without magnesium or calcium and subsequently washed 2 x for 5 minutes. Samples from each litter were pooled and trypsinized in DPBS containing 0.25% trypsin for 10, 15, and 30 min at room temperature for E12, P0, and P9, respectively. Following three washes with fresh DPBS, the tissue was triturated with three progressively smaller (1, 0.5, 0.1 mm) bore polished and sterile pipettes in DPBS containing 250 U/ml DNase, 0.25% glucose, and 8 mg/ml BSA. The triturated cells were diluted 1:4 with cold DPBS and passed through a cell strainer (40 µm mesh) to remove large cellular debris. The cells were collected by mild centrifugation, washed in fresh DPBS and counted. The cells were split into 100,000 cell aliquots, pelleted and snap frozen using liquid nitrogen. Cell pellets were stored at –80 °C.

### Histone chromatin immunoprecipitation
We performed native chromatin immunoprecipitation (ChIP) using validated antibodies against H3K4me1 and H3K27ac according to previously established protocols by the International Human Epigenomics Consortium (IHEC) (*Lorzadeh et al., 2017*). Briefly, cells were lysed in mild non-ionic detergents (0.1% Triton X-100 and Deoxycholate) and protease inhibitor cocktail (Calbiochem) to preserve the integrity of histones harbouring epitopes of interest during cell lysis. Cells were digested by Micrococal nuclease (MNase) at room temperature for 5 min and 0.25 mM EDTA was used to stop the reaction. Antibodies to H3K4me1 (Diagenode: Catalogue #C15410037, lot A1657D) and H3K27ac (Hiroshi Kimura, Cell Biology Unit, Tokyo Institute of Technology) were incubated with anti-IgA magnetic beads (Dynabeads from Invitrogen) for 2 hr. Digested chromatin was incubated with magnetic beads alone for 1.5 hr. Digested chromatin was separated from the beads and incubated with antibody-bead complex overnight in immunoprecipitation buffer (20 mM Tris-HCl pH 7.5, 2 mM EDTA, 150 mM NaCl, 0.1% Triton X-100, 0.1% Deoxycholate). The resulting immunoprecipitations were washed twice by low salt (20 mM Tris-HCl pH 8.0, 2 mM EDTA, 150 mM NaCl, 1% Triton X-100, 0.1% SDS) and then high salt (20 mM Tris-HCl pH 8.0, 2 mM EDTA, 500 mM NaCl, 1% Triton X-100, 0.1% SDS) wash buffers. Immunoprecipitations were eluted in an elution buffer (1% SDS, 100 mM Sodium Bicarbonate) for 1.5 hr at 65 °C. Remaining histones were digested by Protease (Invitrogen) for 30 min at 50 °C and DNA fragments were purified using Ampure XP beads (Beckman Coulter). The library preparation was conducted by Diagenode ChIP-seq/ChIP-qPCR Profiling service (Diagenode Cat# G02010000) using the MicroPlex Library Preparation Kit v2 (Diagenode Cat. C05010013). 50 bp

single-end sequencing was performed on all libraries by Diagenode (Belgium) on an Illumina HiSeq 3000 platform. Two independent biological replicates were performed for each antibody and developmental time point.

## Histone modification ChIP-seq data processing

The sequencing data were uploaded to the Galaxy web platform (usegalaxy.org) for analyses (*Afgan et al., 2016*). 50 bp single-end ChIP-seq reads were aligned to the NCBI37/mm9 reference genome and converted to binary alignment/map (BAM) format by Bowtie2 v.2.3.4 (*Langmead et al., 2009*) with default parameters. Duplicate reads were marked using Picard v.1.52. Quality control metrics after alignment, which were used to evaluate immunoprecipitation sensitivity, were calculated using ChIPQC v.4.2 (*Carroll et al., 2014*). Peak enrichment was computed using MACS v.2.1.1 (*Zhang et al., 2008*) with a false discovery rate (FDR) cutoff of 0.01 (*P*-value <1E-5) using input samples as a control for each replicate. bigWigs were generated and normalized by the total number of mapped reads using the BamCompare and profiles were generated from these bigWigs by calculating average coverage in 50 bp bins using Deeptools v.3.3 (*Ramírez et al., 2016*) for downstream analysis and visualization.

## Identification of active cerebellar enhancers

We first determined consensus peaks between replicates for both H3K27ac and H3K4me1 peaks collected at E12, P0, and P9 using the *intersect* function from Bedtools v.2.28 (*Quinlan and Hall, 2010*). Robust active cerebellar enhancers were identified by overlapping replicated H3K27ac and H3K4me1 peaks called for E12, P0, and P9 samples. The genomic coordinates of the H3K27ac peaks that overlapped with H3K4me1 enriched regions at the same age were used for our list of robust active cerebellar enhancers. We then removed peaks found within promoter sequences by eliminating any peaks found 500 bp upstream or downstream of transcription start sites (TSSs) in the developing cerebellum as determined previously (*Zhang et al., 2018*). The resulting list of robust active cerebellar enhancer sequences at E12, P0, and P9 were used for downstream analysis.

For the comparative analysis with cerebellar postnatal enhancers previously published by *Frank et al., 2015*, H3K27ac and DNase-seq peak coordinates were downloaded from Gene Expression Omnibus (GEO) (GSE60731). The following sequences were downloaded from public enhancer databases: (1) enhancers downloaded from the VISTA Enhancer Browser (https://enhancer.lbl.gov/) (*Visel et al., 2007*) with hindbrain activity were filtered using the 'Advanced Search' tool, selecting 'hindbrain' under Expression Pattern and retrieving only mouse sequences with positive signal and (2) mouse cerebellar neonate enhancer coordinates were downloaded from the Enhancer Atlas 2.0 repository (http://www.enhanceratlas.org/downloadv2.php) (*Gao and Qian, 2020*). For the comparisons, sequences were overlapped with our robust cerebellar enhancer peaks and H3K27ac peaks at E12, P0 and/or P9 using Bedtools v.2.28 (*Quinlan and Hall, 2010*).

## Differential binding analysis

Aligned read counts (BAM file format) from our H3K27ac ChIP-seq experiments mapped to our robust active cerebellar enhancers for E12, P0, and P9 samples were used as input to the package DiffBind v1.4.2 (*Stark and Brown, 2011*). Read counting at each genomic location was conducted, which was subsequently normalized by experimental input samples. The result of counting is a binding affinity matrix containing normalized read counts for every sample at each robust active cerebellar enhancer. For differential binding affinity analysis, three contrasts were set up in DiffBind: E12 vs P0, E12 vs P9, and P0 vs P9. Differential binding was determined by DiffBind using a negative binomial test at an FDR <0.05 threshold. The FDRs and normalized signal difference for each contrast were plotted using the EnhancedVolcano package in R (*Blighe et al., 2020*).

## Temporal classification of cerebellar enhancers

To determine cerebellar enhancers with embryonic-specific activity, H3K27ac signal at E12 was compared to P0 and P9. Enhancers with significantly higher signal at E12 for either contrast were considered 'Early' active enhancers. A region found to be enriched for both contrasts was counted as one enhancer. To determine cerebellar enhancers with postnatal-specific activity, H3K27ac signal at P9 was compared to E12 and P0. Enhancers with significantly higher signal at P9 for either contrast were

considered 'Late' active enhancers. A region found to be enriched for both contrasts was counted as one enhancer. To determine cerebellar enhancers with activity specific to birth, H3K27ac signal at P0 was compared to P9 and E12. Enhancers with significantly higher signal at P0 in both contrasts would identify enhancers that peaked in activity at P0. We did not identify any enhancers that peaked in activity at P0 and conducted the remaining analysis for only Early and Late enhancers.

## Comparison with ENCODE developing mouse brain datasets

Cerebellar H3K4me1 and H3K27ac peaks and robust cerebellar enhancer coordinates were compared to ENCODE histone profiles quantified in the developing mouse hindbrain, midbrain and forebrain (*Gorkin et al., 2020*). ENCODE H3K4me1 and H3K27ac Bigwig and BED files for E12, P0 and Adult samples were downloaded from the ENCODE data Collection and Coordination website (https://www.encodedcc.org) using the experiment IDs listed in the supplementary information provided by the study. Cerebellar histone peaks were overlapped with hindbrain, midbrain and forebrain peaks using the intersect function from Bedtools v2.30.0 (*Quinlan and Hall, 2010*). H3K27ac ChIP-seq signal was then examined for Early and Late robust cerebellar enhancers in hindbrain, midbrain and forebrain samples. This was calculated using the 'multiBigwigSummary' function from DeepTools v.3.3 (*Ramírez et al., 2016*).

## Comparative analysis with developing cerebellar cis-regulatory sequences (CREs) identified by snATAC-seq

Robust cerebellar enhancers were compared to CREs identified previously in the developing cerebellum by snATAC-seq (*Sarropoulos et al., 2021*). Datasets containing CRE coordinates, global CRE clusters and associated information about peaks and cells were downloaded from https://apps.kaessmannlab.org/mouse_cereb_atac/. The CRE coordinates and global CRE clusters were isolated from the file '*Mouse_Cereb_ATAC_CRE_info.txt*'. CRE locations were overlapped with robust cerebellar enhancer genomic coordinates using the 'intersect' function from Bedtools v2.30.0 (*Quinlan and Hall, 2010*). Robust cerebellar enhancers were assigned a cell-type based on the cell-type in which an overlapping CRE was detected. CREs were previously clustered into 26 clusters based on activity using an iterative clustering procedure (*Sarropoulos et al., 2021*). Average chromatin accessibility signal for each cell type at all available developmental stages were isolated from the file '*Mouse_Cereb_ATAC_CREs_SE.rds*' using a custom script (Source Code File 1) provided by the corresponding authors. The CREs were filtered for those that overlapped with Early and Late active robust cerebellar enhancers and ATAC-seq signal was plotted for progenitors, granule cells, Purkinje cells, and interneurons.

## Transcription factor motif enrichment analysis

Transcription factor motif enrichment was calculated using the software HOMER using the script FindMotifsGenome.pl with default parameters (*Heinz et al., 2010*). Analyses for Early and Late active enhancers were conducted separately. Motif enrichment was statistically analyzed using a cumulative binomial distribution. Enriched motifs were aligned with known transcription factor binding sites to determine the best matches. Top known motif matches were filtered based on expression within the developing cerebellum at E12 for 'Early' active enhancers and P9 for 'Late' active enhancers.

## Cerebellar enhancer target gene prediction and co-expression analysis

To identify possible gene targets of our robust cerebellar enhancers, the correlation between H3K27ac signal and mRNA expression of genes located in cis at E12, P0, and P9 was calculated. For a given enhancer, a gene located in cis was considered a possible target if it was positively correlated with H3K27ac signal throughout time. These genes were then filtered based on location using conserved topologically associating domains (TADs), which are areas of the genome that preferentially interact (*Dixon et al., 2012*). These TADs are conserved between different cell types and even across species and were established using Hi-C data generated, previously. Gene target candidates for a given enhancer were curated for those located within the same TAD. Predicted gene targets were then ranked based on their Pearson correlation coefficient value. For the predicted gene targets of Early and Late active enhancers, we conducted *k-means* clustering of predicted gene targets separately. Input for this analysis was gene expression captured from cerebella at 12 embryonic and postnatal time points (*Ha et al., 2019*). Briefly, gene expression was quantified using Cap Analysis of Gene

Expression followed by sequencing (CAGE-seq) for mouse cerebellar samples dissected every 24 hr from E11-P0 and then every 72 hr until P9 (12 in vivo time points in total). The number of clusters for the *k-means* clustering was determined using the Elbow analysis for each classified group of enhancers: Early (n=4) and Late (n=4).

A permutation analysis was conducted to evaluate whether cerebellar genes were enriched in robust cerebellar enhancer putative target genes. We generated 10,000 permutations of 2261 (total number of putative targets) randomly selected from all expressed genes in the developing cerebellum (~30,000 genes) (*Ha et al., 2019*). The number of cerebellar genes were then assessed in each permutation. One-sided p-values were calculated by dividing the number permutations containing a given number cerebellar genes by the total number of permutations (10,000).

## Expression analysis of mouse and human single-cell RNAseq (scRNA-seq) datasets

Pax3 and Bhlhe22 expression values were analyzed in scRNA-seq datasets generated in the developing mouse and human developing cerebellum (*Carter et al., 2018*; *Aldinger et al., 2021*). The mouse cerebellar dataset was received as an 'h5' file from the corresponding authors, containing normalized expression values and annotations for all cells analyzed. Average normalized expression values were then calculated for 18 cell-type clusters previously determined using cell-type-specific markers (*Carter et al., 2018*). For the human cerebellar scRNA-seq dataset, we received a dataset from the corresponding author containing average normalized expression values for the 21 cell-type clusters previously determined using cell-type-specific markers (*Aldinger et al., 2021*). To evaluate Pax3, Bhlhe22, and Cdc42bpb expression in the cell types of the mouse and human developing cerebellum, normalized expression values for each cell-type cluster were plotted in bar and radar plots.

## Tissue preparation for histology

Embryos harvested between E12.5 to E15.5 were fixed by immersion in 4% paraformaldehyde in 0.1 M phosphate buffer (PB, pH 7.4) for 1 hr at 4 °C. Postnatal mice between P0.5 to P9.5 were perfused through the heart with a saline solution followed by 4% paraformaldehyde/0.1 M PBS. The brain tissues were isolated and further fixed in 4% paraformaldehyde in 0.1 M PB for 1 hr at room temperature. Fixed tissues were rinsed with PBS, followed by cryoprotection with 30% sucrose/PBS overnight at 4 °C before embedding in the Optimal Cutting Temperature compound (Tissue-Tek). Tissues were sectioned at 12 μm for immunofluorescence experiments and cryosections were mounted on Superfrost slides (Thermo Fisher Scientific), air dried at room temperature, and stored at −80 °C until used. Sagittal sections were cut from one side of the cerebellum to the other (left to right, or vice versa). In all cases, observations were based on a minimum of 3 embryos per genotype per experiment.

## Cerebellar immunostaining

Tissue sections were first rehydrated in PBS (3x5 minute washes) followed by a phosphate buffered saline with Triton X-100 (PBS-T) rinse. Sections were then incubated at room temperature for 1 hr with blocking solution (0.3% BSA, 10% normal goat serum, 0.02% sodium azide in PBS-T). Following the blocking step, the slides were incubated with primary antibody in incubation buffer (0.3% BSA, 5% normal goat serum, 0.02% sodium azide in PBS-T) at room temperature overnight in a humid chamber. Following the overnight incubation, the slides were rinsed in 3x10 min PBS-T washes. The sections were then incubated with the appropriate secondary antibody at room temperature for 1 hr, followed by three 0.1 M PB washes and one 0.01 M PB wash. Coverslips were applied to the slides using Fluor-Save mounting medium (345789, Calbiochem). The primary antibodies used were: rabbit anti-Bhlhe22 (1:1000, a gift from Dr. Michael Greenburg, Harvard University), mouse anti-Neurod1 (1:500, Abcam, ab60704), mouse anti-Pax3 (1:500, R&D systems, MAB2457), rabbit anti-Pax2 (1:200, Invitrogen, 71–6000), mouse anti-NeuN (1:100, Millipore, MAB377), rabbit anti-Calbindin (1:1000, Millipore, AB1778), rabbit anti-Foxp2 (1:2000, Novus Biologicals NB100-55411), chicken anti-Doublecortin: (1:100, Abcam ab153668). For immunofluorescence, secondary antibodies (Invitrogen) labeled with fluorochrome were used to recognize the primary antibodies.

## Granule cell culture

Granule cells were isolated and cultured as previously described (*Lee et al., 2009*). Briefly, cerebella from litters of P6 mice were pooled and digested at 37 °C for 20 min in 10 U ml−1 papain (Worthington), and 250 U ml−1 DNase in EBSS using the Papain Dissociation Kit (Worthington, Cat #:LK003150). The tissue was mechanically triturated and suspended cells were isolated and resuspended in EBSS with albumin-ovomucoid inhibitor solution. Cell debris was removed using a discontinuous density gradient and cells were resuspended in HBSS, glucose and DNase. The cell suspension was then passed through a 40 µm cell strainer (Falcon 2340), layered on a step gradient of 35% and 65% Percoll (Sigma), and centrifuged at 2500 rpm for 12 min at 25 °C. Granule cells were harvested from the 35/65% interface and washed in HBSS-glucose. Granule cells were then resuspended in Neurobasal medium and 10% FBS and pre-plated on lightly coated poly-D-lysine-coated dishes for 20 min. This step allows any heavier cells to drop and adhere to the coated surface while the granule cells are retained in the media. Granule cells in the media were then collected, washed and counted using a Hemocytomoter. The cells were then plated on 25 mm or 12 mm poly-D-lysine (Sigma), laminin-coated coverglasses placed in six-well plates with Neurobasal medium containing B-27 serum-free supplement, 2 mm l-glutamine, 100 U/ml penicillin, and 100 µg/ml streptomycin (pen-strep; Invitrogen, Grand Island, NY) and 0.45% d-glucose (Gibco). Granule cells were incubated at 37 °C at 5% $CO_2$ were incubated for 3 days in vitro (DIV).

For aggregate cultures, aggregates were allowed to form by incubating purified granule cells for 20 hr on uncoated tissue culture dishes in DMEM containing 10% FBS, 0.45% D-glucose, Pen-strep, 2 mM L-glutamine at 4E6 cells/ml. Aggregates were then washed and cultured in Neurobasal/B27 medium on poly-d-lysine/laminin-treated chamber slides at 37 °C/5% $CO_2$. Neuronal processes extend from aggregates and most form neurite bundles. After several hours, small bipolar granule cells migrate unidirectionally away from the cell clusters along these neurites and neurite bundles by extension of processes, followed by translocation of cell bodies outside of the aggregate cell cluster margin. For immunofluorescence experiments, cells were fixed in 4% paraformaldehyde for 10 min and washed with calcium and magnesium-free PBS.

## RNA interference

For the knockdown of Bhlhe22, we purchased ON-TARGETplus SMARTPool Mouse Bhlhe22 siRNA from Horizon Discovery (Cat ID: L-063262–01). Control samples were transfected with ON-TARGETplus Non-targeting Control Pool (Cat ID: D-001810–10). siRNA molecules were electroporated into isolated postnatal cerebellar granule cells using the Nucleofector 2b Device (Lonza, AAB-1001) as previously described (*Gärtner et al., 2006*). Briefly, after cells were isolated (described above), 6-7E6 cells were resuspended in nucleofection solution and mixed with 3 µg of pCAG-EGFP plasmid (Addgene, 89684) and 600 nM of siRNA. Cuvettes were loaded with cellular solution and nucleofected using program O-03. After electroporation, cells were allowed to recover in DMEM media in a humidified 37 °C/5% $CO_2$ incubator for 90 min. Cells were washed and resuspended in either culture media for plating (dissociated cultures) or DMEM media for overnight incubation (aggregate cultures).

## RNA isolation and reverse transcription followed by quantitative PCR (RT-qPCR)

RNA was collected from cultured granule cells using the Monarch Total RNA Miniprep Kit (NEB). Then cDNA was reverse transcribed using SuperScript IV First-Strand Synthesis System (Invitrogen) using random hexamers. Quantitative PCR was conducted using the Applied Biosystems Fast SYBR Green Master Mix reagent and Applied Biosystems 7500 Real-time PCR system. PCR conditions were as follows: 95 °C for 20 seconds, 40 cycles of 95 °C for 3 s, and 60 °C for 30 s followed by 95 °C for 15 s, 60 °C for 1 min, 95 °C for 15 s and 60 °C for 15 s. Three biological replicates were analyzed for each target gene. Amplification of eGFP was used as a reference gene to normalize the relative amounts of successfully transfected cells between treated and control experiments. Gene specific primers are listed in *Supplementary file 9*. Expression profiles for each gene were calculated using the average relative quantity of the sample using the deltaCT method (*Livak and Schmittgen, 2001*). For comparisons between siRNA treated and control samples, means were compared using a two-tailed t-test. Results were expressed as the average ± SE, and p-values <0.05 were considered significant.

## Image analysis and microscopy

Analysis and photomicroscopy were performed using a Zeiss Axiovert 200 M microscope with the Axiocam/Axiovision hardware-software components (Carl Zeiss) and downstream image analysis was conducted using the AxioVision software v.4.9.1 (Carl Zeiss).

For cerebellar granule cell aggregate cultures, aggregate size determined using the tracing tool and all aggregates analyzed were within 1000 µm$^2$ of each other. Transfected cells were identified by examining eGFP expression and for each biological replicate/experimental treatment, 20 aggregates were examined. Granule cell migration was measured by calculating the distance of migrated cells from the edge of the aggregate on captured images. Mean migration distance was calculated for each aggregate, and the average of all 20 aggregates was used for statistical analysis. The distribution of migrated cells from the aggregate was calculated for the following ranges:<25 µm, 25–50 µm, 50–75 µm, 75–100 µm, >100 µm. For each range, the average percentage was calculated for 20 aggregates per replicate. For comparisons between siRNA treated and control samples, means were compared using a two-tailed t-test. Results were expressed as the average ±SE, and p-values <0.05 were considered significant.

For the quantification of Pax3 and Bhlhe22 expressing cells, images captured for immunofluorescent staining were used for cell counting. Three randomly selected 100 µm square portions of each sagittal section of the developing cerebellum were analyzed at the indicated developmental stages. The percentage Pax3 and Bhlhe22 expressing cells that colocalized with cell type markers was determined by counting the number of co-stained cells dividing the total by the number of Pax3 or Bhlhe22 expressing cells in a square. Three square portions were counted for three sagittal sections for each of the three biological replicates. Each data point on the respective bar graphs represent an average calculated for each sagittal section.

## Plots and statistical methods

All plots and correlation analysis were generated in R version 3.2.3 and figures were produced using the package ggplot2. Bedtools v.2.28 (*Quinlan and Hall, 2010*) was used for comparing and overlapping the genomic coordinates of peaks and existing genomic features described in the manuscript. Gene Ontology enrichment analyses were conducted using the clusterProfiler package in R (*Yu et al., 2012*). Boxplots represent the median (centre line), first and third quartiles (top and bottom of box, respectively) and confidence intervals (95%; black lines). Genome browser screenshots were taken from the IGV genome browser (*Robinson et al., 2011*). Bar plots results were expressed as the average and the corresponding error bars represent standard error.

## GWAS SNP enrichment analysis

The NCBI LiftOver tool was used to convert the coordinates of cerebellar enhancers from mm9 to hg19 to hg38, and BEDTools (*Quinlan and Hall, 2010*). 6630/7024 robust cerebellar enhancers were converted from mm9 to hg38. Single-nucleotide polymorphisms (SNPs) were retrieved from the GWAS Catalog (*Buniello et al., 2019*) downloaded on March 8th, 2020. The SNPs were then filtered by their associated traits. Traits containing the word 'autism' were selected and from this list any traits containing the word 'or' were excluded. This resulted in a final list of 8 traits (*Supplementary file 10*) and the associated SNPs were used as input for our analysis. The software Genomic Regulatory Elements and Gwas Overlap algoRithm (GREGOR) v.1.4.0 (*Schmidt et al., 2015*), a tool to test for enrichment of an input list of trait-associated index SNPs in experimentally annotated regulatory domains, was used to identify enrichment of trait-specific disease variants within enhancers. An underlying hypothesis of GREGOR is that both trait-associated SNPs and variants in strong linkage disequilibrium (LD) may be deemed as causal. For this, we used the European population reference file (EUR; LD window size = 1 Mb; LD r$^2$ ≥0.7) from 1000 G data (Release date: May 21, 2011). The probability of an overlap of either a SNP or at least one of its LD proxies with our enhancers relative to a set of matched control variants was used to evaluate significance of overlap. The enrichment p-value is the probability that the overlap of control variants with our enhancers is greater than or equal to the overlap of the GWAS variants with converted robust cerebellar enhancers.

## De novo mutation analysis

De novo mutations were detected using whole-genome sequencing data from the MSSNG (*Yuen et al., 2016*) and Simons Simplex Collection (SSC) (*Isoda et al., 2017*) cohorts using a pipeline involving DeNovoGear (*Ramu et al., 2013*) as previously described (*C Yuen et al., 2017*). To maximize data homogeneity, we included only individuals sequenced on the Illumina HiSeq X platform. Individuals having a total DNM count more than three standard deviations above the mean of the cohort were excluded. The NCBI LiftOver tool was used to convert the coordinates of cerebellar enhancers from mm9 to hg19 to hg38, and BEDTools (*Quinlan and Hall, 2010*) was used to identify DNMs overlapping these coordinates. Contingency tables (2x2) were generated containing counts of the number of DNMs in ASD-affected individuals and unaffected siblings either overlapping or not overlapping each dataset (cerebellar enhancer or H3K27ac peak coordinates). Fisher's exact test was used to determine statistical significance. Copy number variants (CNVs) ≥ 1000 bp were detected from the MSSNG and SSC WGS data using a pipeline involving the algorithms ERDS (*Zhu et al., 2012*) and CNVnator (*Abyzov et al., 2011*) as previously described (*Trost et al., 2018*). A CNV was deemed to be de novo if it was detected by both ERDS and CNVnator in the child but by neither algorithm in both parents. We then used BEDtools (*Quinlan and Hall, 2010*) to identify de novo CNVs overlapping our cerebellar enhancers.

# Additional information

### Competing interests

FANTOM 5 Consortium: The other authors declare that no competing interests exist.

### Funding

| Funder | Grant reference number | Author |
|---|---|---|
| NSERC Discovery Award | | Daniel Goldowitz |

The funders had no role in study design, data collection and interpretation, or the decision to submit the work for publication.

### Author contributions

Miguel Ramirez, Conceptualization, Data curation, Software, Formal analysis, Validation, Investigation, Visualization, Methodology, Writing - original draft, Project administration, Writing - review and editing; Yuliya Badayeva, Data curation, Software, Formal analysis, Validation, Investigation, Visualization, Methodology, Writing - original draft, Writing - review and editing; Joanna Yeung, Formal analysis, Supervision, Validation, Methodology, Project administration; Joshua Wu, Formal analysis, Validation, Investigation, Visualization, Methodology, Writing - review and editing; Ayasha Abdalla-Wyse, Data curation, Software, Visualization; Erin Yang, Validation, Investigation, Visualization, Methodology; FANTOM 5 Consortium, Data curation, Resources; Brett Trost, Resources, Data curation, Software, Formal analysis, Validation, Investigation, Visualization, Methodology, Writing - original draft, Writing - review and editing; Stephen W Scherer, Resources, Data curation, Supervision, Funding acquisition, Investigation, Writing - original draft, Writing - review and editing; Daniel Goldowitz, Conceptualization, Resources, Data curation, Supervision, Funding acquisition, Investigation, Methodology, Writing - original draft, Project administration, Writing - review and editing

### Author ORCIDs

Miguel Ramirez http://orcid.org/0000-0001-6315-4291
Joanna Yeung http://orcid.org/0000-0003-0551-5305
Erin Yang http://orcid.org/0000-0001-5629-2362
Brett Trost http://orcid.org/0000-0003-4863-7273
Stephen W Scherer http://orcid.org/0000-0002-8326-1999
Daniel Goldowitz http://orcid.org/0000-0003-4756-4017

### Ethics

All studies were conducted according to the protocols approved by the Institutional Animal Care and Use Committee and the Canadian Council on Animal Care at the University of British Columbia.

### Decision letter and Author response

Decision letter https://doi.org/10.7554/eLife.74207.sa1
Author response https://doi.org/10.7554/eLife.74207.sa2

## Additional files

### Supplementary files

• Supplementary file 1. Coordinates in BED file format of all 7024 robust cerebellar enhancers.

• Supplementary file 2. Coordinates of Early and Late active robust cerebellar enhancers.

• Supplementary file 3. Quality control metrics evaluating H3K27ac and H3K4me1 ChIP-seq sensitivity. Map%: Percentage reads mapped, Filt%: percentage of reads filtered, Dup%: duplication rate, ReadL: read length, FragL:, FRiP%: fraction of reads in peaks, RSC: relative strand correlation.

• Supplementary file 4. TF motifs enriched in robust cerebellar enhancers. This table contains the motif symbol, the TF in the JASPAR database that best matches the enriched motif, protein family of the best match TF, percentage of robust cerebellar enhancers and background sequences containing each motif.

• Supplementary file 5. List of the most highly correlated putative gene targets of robust cerebellar enhancers. This file contains coordinates of the enhancer, gene target symbol, and Pearson correlation coefficient.

• Supplementary file 6. Robust cerebellar enhancers containing a Pax3 motif and their putative target genes. The rightmost column indicates whether the putative target gene has been previously implicated in cerebellar development.

• Supplementary file 7. Coordinates of mouse robust cerebellar enhancers and putative target genes converted to human genome build hg38. The rightmost column ('Orthologous') indicates whether an enhancer and it's putative target genes are found on the same chromosome after conversion.

• Supplementary file 8. Gene targets for enhancers enriched with ASD variants from the GREGOR analysis. The second column indicates the group to which the gene belongs to in the SFARI gene database of ASD candidate genes. S: Syndromic gene category; 1: Category 1 (High Confidence); 2: Category 2 (Strong Candidate).

• Supplementary file 9. Primers used for RT-qPCR analysis.

• Supplementary file 10. GWAS Catalog traits used to identify variants associated with ASD. All variants associated with these traits were used as input for the ASD variant enrichment analysis conducted using GREGOR.

• Transparent reporting form

• Source code 1. Generate psuedobulk expression values (CPM) by developmental stage.

### Data availability

Sequencing data have been deposited in GEO under accession code: GSE183697.

The following dataset was generated:

| Author(s) | Year | Dataset title | Dataset URL | Database and Identifier |
|---|---|---|---|---|
| Ramirez M, Badayeva Y, Yeung J, Wu J, Yang E | 2021 | Temporal analysis of enhancers during mouse brain development reveals dynamic regulatory function and identifies novel regulators of cerebellar development | https://www.ncbi.nlm.nih.gov/geo/query/acc.cgi?acc=GSE183697 | NCBI Gene Expression Omnibus, GSE183697 |

The following previously published datasets were used:

| Author(s) | Year | Dataset title | Dataset URL | Database and Identifier |
|---|---|---|---|---|
| Frank CL, Liu F, Wijayatunge R, Song L, Biegler MT, Yang MG, Vockley CM, Safi A, Gersbach CA, Crawford GE, West AE | 2015 | Regulation of chromatin accessibility and Zic binding at enhancers in the developing cerebellum | https://www.ncbi.nlm.nih.gov/geo/query/acc.cgi?acc=GSE60731 | NCBI Gene Expression Omnibus, GSE60731 |
| Klisch TJ, Xi Y, Flora A, Wang L, Li W, Zoghbi HY | 2011 | The Atoh1 targetome in murine postnatal cerebellum | https://www.ncbi.nlm.nih.gov/geo/query/acc.cgi?acc=GSE22111 | NCBI Gene Expression Omnibus, GSE22111 |
| Aldinger KA | 2021 | Spatial and cell type transcriptional landscape of human cerebellar development | https://www.covid19cellatlas.org/aldinger20 | Human Cell Atlas, aldinger20 |
| Carter RA | 2018 | A Single-Cell Transcriptional Atlas of the Developing Murine Cerebellum | https://www.ebi.ac.uk/ena/browser/view/PRJEB23051?show=reads | European Nucleotide Archive, PRJEB23051 |
| Sarropoulos I | 2021 | Developmental and evolutionary dynamics of cis-regulatory elements in mouse cerebellar cells | https://apps.kaessmannlab.org/mouse_cereb_atac/ | Developing mouse cerebellum snATAC-seq atlas, kaessmannlab |

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
