## [Editor Report]

This manuscript is a valuable study to understand regulatory elements in the early developing mouse cerebellum. The authors have generated important ChIP-seq datasets from embryonic and early postnatal mouse cerebellum. The authors use these data to convincingly identify enhancers in the developing cerebellum. The authors have also made their data and analyses available online in the "Developing Mouse Cerebellum Enhancer Atlas."

---

## [Decision Letter]

**Decision letter after peer review:**

Thank you for resubmitting your work entitled "Temporal analysis of enhancers during mouse brain development reveals dynamic regulatory function and identifies novel regulators of cerebellar development" for further consideration by *eLife*. Your revised article has been evaluated by Kathryn Cheah (Senior Editor) and a Reviewing Editor.

The manuscript has been improved but there are some remaining issues that need to be addressed, as outlined below:

Importantly, we think this manuscript would fit a "Resource" category better than its current status as a research article. However, even to meet that bar we believe there are substantial revisions that need to be made. The reviewers felt that while the data might constitute a valuable resource for specific researchers that the novelty and impact of the dataset overall was not sufficient for a research article. However, if the authors can integrate previously published datasets more effectively and make your dataset more easily digestible/accessible by a broader neuroscience audience, we believe this could be a valuable dataset for the community.

*Reviewer #1:*

This manuscript aims to identify and understand regulatory elements in the early developing mouse cerebellum. To this end, the authors generate datasets of H3K4me1 and H3K27ac ChIP-seq in embryonic and early postnatal mouse cerebellum. The authors use these data to identify potential enhancers and carry out a number of confirmatory experiments that overlap their data with previously published ones as well as showing expression patterns of target genes that fit with the enhancer data.

The strengths of the manuscript include:

1) the generation of embryonic data that have been missing in the literature.

2) the integration with previously published datasets.

3) the IHC/ISH experiments that correlate expression with enhancer activity.

4) the functional characterization of Bhlhe22, which had not previously been described in the literature.

The weaknesses of the manuscript are:

1) The bulk approach that could potentially mask patterns driven by rare cell types.

2) Lack of mutational testing of specific enhancers.

In general, I think the manuscript is well written and easy to understand. The authors make a case for the need and novelty of their datasets. They also try to bolster their findings using a number of additional wet-bench approaches as well as integration with published datasets.

The biggest weakness is the lack of cell-type resolution presented using this bulk approach. The authors do a reasonable job trying to address this with the data in hand and mention in the discussion that finer resolution is needed. I'm just not sure that given the current state of what is possible technically that the current approach provides sufficiently important data (although contribution of embryonic data is always needed).

*Reviewer #2:*

The authors here sought to identify enhancers that are active in developing cerebellum and distinguish early (e.g. embryonic) from late (e.g. early postnatal) activity. To accomplish this, the authors performed new ChIP-seq experiments on two histone modifications that differentiate poised/latent from active enhancers, H3K4me1 and H3K27ac respectively, interrogating E12, P0, and P9 mouse cerebellum. The authors performed differential binding analysis and intersected these epigenomic datasets with published transcriptomic and autism human genetics datasets. While the scope is limited, the datasets and analysis are of sufficient rigor and reproducibility and the authors validate two genes with novel cerebellar function, thus the results and resulting resource of annotated enhancers should be sound.

The main weakness with this study is limited novelty and impact due to studies that have generated similar datasets and general findings. Other studies have generated epigenomics datasets for developing mouse cerebellum, including recently with single cell resolution using snATAC-seq, and more detailed time courses and complete chromatin signatures. The datasets here are largely redundant with work from the mouse ENCODE project, which profiled hindbrain across eight developmental stages that overlap with the two early timepoints here, as well as adult cerebellum. Similarly, the general concept of enhancers anchoring dynamic developmental processes, including in brain and specifically cerebellum, is now firmly established. Nonetheless, focused studies such as this one have value in that the results and datasets are digestible, curated, and directed toward a specific audience that are likely to appreciate the findings and use the results as a resource. The value of the work is the focus on identification and characterization of enhancers and their target genes with differential early developmental versus post-natal activity in the cerebellum.

While this focused study should have an audience, it would be stronger if the authors interfaced their data and results with other larger and more granular (e.g. single cell or more chromatin marks) datasets. If this is done effectively, this manuscript will have clear stand alone value as a complement to the larger, but not curated or cerebellum focused ENCODE data. And would link to the emergine single cell datasets that also focus on dynamic transcriptional and epigenomic changes. This would put the results from this work in the context of the larger and centralized efforts, while maintaining the useful focus.

Analysis and results:

1. The IHC in the manuscript lacks any quantitation and co-labeling/co-expression patterns are not clear with the lower magnification images. Figure 3 and 6 (and supp fig) analysis should have a quantitative summary of staining and of co-labeling. For example, Figure 3 is used to argue for presence or absence of co-expression/co-labeling between Pax3 and Ptf1a, Pax2, and Foxp2. I would like to see neuroanatomical annotation lines as overlay on sections for non-experts. Finally, I think higher magnification insets that clearly show co-labeling (or lack thereof) would make these figures stronger.

2. The P0 enhancer set is much smaller than the E12 or P9, raising questions about technical sensitivity. The authors make the conclusion that there are no P0 enriched enhancers, but this could also simply be a lack of sensitivity. The authors could address this by presenting information regarding the signal to noise and comparing their data to published ENCODE P0 hindbrain data.

3. The authors should compare their datasets to the mouse ENCODE developmental hindbrain datasets. Ideally, they would verify the patterns they found, showing that their datasets are high quality and correlated with the ENCODE data and then also review the reproducibility of the early vs. late enhancers. The authors can also use the P9 data to compare to the adult cerebellum in mouse ENCODE to see how similar these datasets are – it would be of interest if P9 is capturing novel enhancers. The authors could also look at dynamic chromatin accessibility to see if the H3K27ac + H3K4me1 signature here is more sensitive to active enhancers than accessibility alone (which would be expected and would further justify the value of the data here).

4. It would be of interest to see how the enhancer sets identified here map to the recently published snATAC-seq datasets from mouse cerebellum across development. This publication was referenced in the manuscript, but not described – Sarropoulos Science 2021. More specifically, can the early and late enhancers here be assigned to shared or distinct cerebellar cell types? Also as above, it would add value to this study if the H3K27ac+H4K3me1 activity signatures produces improved sampling of active enhancers vs. the snATAC.

5. The authors could intersect their target early and late genes with recent snRNA-seq data from developing human cerebellum (Aldinger Nat Neuro 2021).

6. Some general issues with the target gene set analsyis:. For the k-means clustering of target genes presented in Figures 4 and 5, the authors should provide rationale for using the k value selected. K-means is a great algorithm for this due to simplicity and wide use, but can require optimization and nested approaches. What other values of k were used? Did a nested approach reveal any subclusters within the four major groups at either timepoint? It is fine to use some arbitrary/post-hoc selection of best clusters, but the rationale should be described. Figure 5 should include N for genes per cluster. Finally I'm not sure what the GeneRatio measure is supposed to capture. IT would be more straightforward to use an observed vs. expected fold enrichment, as counts can skew depending on number of genes and fold enrichment is standard in GO analysis.

7. Motif analysis: Motifs are often similar for families, so motif to TF matching is imprecise at best. Fine to select best guess TF as authors do, but good to also show motif logo in Figure 3A and include mention that the actual TF could be a different family member unless orthogonal evidence exists.

8. Intersection with ASD variants is flimsy. The authors should use some presumed negative for non-brain and other brain GWAS to show that the ASD SNP findings are relevant e.g. have stronger signal than presumed negative or less relevant phenotypes. The overall enrichment with de novo mutations in enhancesr does not demonistrate any specific cerebellar relevance unless these enhancers are not active in other brain regions. Other studies have showed general enrichment of de novo mutations in brain enhancers. Not sure whether the rare CNV with the cerebellar gene is useful or convincing of relevance as the N is 1 and the gene expression outside the cerebellum is bot shown or discussed. All this analysis is fine, but shouldn't be treated as a major finding. I think the other parts of this study are much stronger, so de-emphasizing this would not hurt the paper.

9. The argument that MGI mouse cerebellum phenotype is evidence for validity needs to be compared to some random set in order to evaluate how strong the 98/2261 finding is- e.g. via permutation analysis selecting random sets of 2261 genes to establish background.

10. Figure 2B is not needed – just put numbers in legend or overlay on volcano plots in 2A.

11. Figure 2C would be more informative with longitudinal data showing change over time.

Text

1. The authors highlight their study as the first to characterize developmentally active enhancers, both in the abstract and discussion. This is an over-statement at best and the authors would do better to both remove this language and to directly deal with some of the other datasets, as described below.

2. The introduction should include brief description of the other relevant studies, particularly the mouse ENCODE data and recent snATAC-seq and explain the novelty of approach here. The discussion should devote space to comparing the results from this study and the mouse ENCOE and snATAC-seq specifically.

3. The last paragraph of the introduction is essentially a detailed summary of the findings. I would prefer this be the first paragraph of the discussion to avoid redundancy and present results before making these claims. A brief general summary statement is fine.

*Reviewer #3:*

Strengths of the manuscript:

The authors have carried out a stringent analysis of genomic enhancers that are dynamically active in specific cell populations over a period of cerebellar development. Their data represents a highly useful resource for the field of cerebellar research.

An enrichment analysis of the enhancers for specific transcription factor binding sites identified potential transcription factors previously not well-known to regulate cerebellar development.

The authors demonstrate how their dataset could be exploited to identify novel genes in cerebellar development (e.g., Bhlhe22) as well as potential disease genes and mechanisms (e.g. in autism spectrum disorder).

Weaknesses:

The authors select three timepoints during cerebellar development for their ChIP-seq analysis. The rational for selecting these specific timepoints is not very clear, especially as they do not cover the entire period of cerebellar development in the mouse.

While the resource itself is very valuable, the presentation of example genes that the authors extract from their analyses (Pax3, Bhlhe22, and Cdc42bpb) is rather short and lacking in detail. For example, the authors could have further strengthened their findings by utilising recent single-cell datasets, in particular the single-cell transcriptional atlas of the developing cerebellum published by Carter et al.

It won't be easy for the scientific community to interact with the generated data.

The authors use in silico tools to identify transcription factor binding sites enriched in the identified active enhancers. No experimental validation is provided for this analysis. This would have been particularly relevant for transcription factors that have not been previously well-described in cerebellar development including Pax3, which the authors present as a novel marker for GABAergic progenitors. Furthermore, it would have been helpful to go into further detail and to provide some predicted candidate genes that might be regulated by Pax3 in GABAergic progenitors.

Figure 3: Very limited expression data is shown for Pax3. It would have been helpful to show additional immunostainings at least for P9 (same timepoint as used for the ChIP-seq analysis) if not for further timepoints during cerebellar development. Moreover, it would have been really interesting to compare this to the data available on Pax3 from Carter et al. The authors mention this in their discussion; it would be helpful to include this data in the relevant figure and also to comment on how expression of Pax3 changes during the course of cerebellar development.

With regards to Pax3, there are some interesting papers that link Pax3 to cerebellar development and cerebellar disorders and that the authors could comment on in their discussion; for example, Kioussi and Gruss (1994) and Jalali et al., (2008), and Zagozewski et al., (2020).

I am not sure that I agree with the authors about the fact that 98 out of 2261 identified target genes result in a cerebellar phenotype when knocked out particularly demonstrates the validity of the high throughput approach (page 12). Is this a significant number?

The use of CAGE-sequencing data in this manuscript is very unclear. The authors refer to a previous manuscript (Zhang et al., 2018); however, in this study reports transcriptomic data from only three timepoints (E13, E15, E18) and not 12 timepoints as mentioned in by the authors? The authors should better describe which data they used. Also, given that the CAGE-Seq data is from bulk-sequencing, it would have been much more informative to look at gene expression of their candidate transcription factors in the Carter et al., single-cell dataset. Furthermore, this would have allowed the authors to look beyond the P9 timepoint and thus for a more complete gene expression analysis over cerebellar development. This is also true for Figures 1E,G.

The approach of using the enhancer data set to identify novel target genes not previously identified in cerebellar development is unclear to me. Using their described filtering approach, the authors describe genes that differentially regulated by Atoh1. Could these not have been identified from the original dataset by Klisch et al., 2001? What does the initial identification of the enhancers add here?

Figure 6: As above, the authors should include analysis of Bhlhe22 expression in the Carter et al., single-cell dataset.

With regards to Bhlhe22, the authors should comment on the protein that is encoded by this gene and its potential function during cerebellar development.

Figure 7 could be much more clearly presented. The font size is really small and the figures not very clear. It would be nice to see successful knockdown of Bhlhe22 by immunostaining.

The authors normalize gene expression to co-transfected GFP. Can the authors demonstrate that cells that are transfected with GFP were also transfected with the siRNA construct?

Can the authors comment on the evolutionary conservation of their identified enhancer elements? This is particularly relevant for their analysis of variants associated with ASD. Are the loci of interest conserved between mouse and human and thus, do enhancer activity and transcription factor occupancy occur at orthologous regions distal to the same gene promoters?

For the identification of CDC42BPB, it would have been nice to link this back to expression data over time and to provide more detail on the potential transcription factors involved.

Finally, given that this is a very important dataset for the scientific community, I wish it would be easier for others to interact with the data rather than just depositing data files into GEO. It would have also been really helpful to provide supplementary tables listing enhancer elements and associated transcription factors as well as nearby genes that the authors have identified. Currently, it is not transparent which lists of genes etc were used for the various analyses shown.

---

## [Author Response]

Reviewer #1:The biggest weakness is the lack of cell-type resolution presented using this bulk approach. The authors do a reasonable job trying to address this with the data in hand and mention in the discussion that finer resolution is needed. I'm just not sure that given the current state of what is possible technically that the current approach provides sufficiently important data (although contribution of embryonic data is always needed).

The bulk tissue dataset that we present provides consensus (i.e. data from many cells) and high confidence signals that point us to coordinates that will serve as benchmarks for validation for past and future single-cell assays. We address the issue of cell-type resolution by comparing our data with recent single cell/nuclei data (Sarrapolous et al., 2021; Carter et al., 2018; Aldinger et al., 2021). The outcome is that we find a significant overlap between our data and the single cell/nuclei data. These analyses and results can be found in a new section in our manuscript entitled ‘Comparison of robust cerebellar enhancers with chromatin accessibility identified in single cells’ (pp. 11) and in the sections that report our findings Pax3 (pp.14-15) and Bhlhe22 (pp.19).

Reviewer #2:The main weakness with this study is limited novelty and impact due to studies that have generated similar datasets and general findings. Other studies have generated epigenomics datasets for developing mouse cerebellum, including recently with single cell resolution using snATAC-seq, and more detailed time courses and complete chromatin signatures. The datasets here are largely redundant with work from the mouse ENCODE project, which profiled hindbrain across eight developmental stages that overlap with the two early timepoints here, as well as adult cerebellum. Similarly, the general concept of enhancers anchoring dynamic developmental processes, including in brain and specifically cerebellum, is now firmly established. Nonetheless, focused studies such as this one have value in that the results and datasets are digestible, curated, and directed toward a specific audience that are likely to appreciate the findings and use the results as a resource. The value of the work is the focus on identification and characterization of enhancers and their target genes with differential early developmental versus post-natal activity in the cerebellum.

We believe that our dataset is not redundant with the chromatin profiles generated from the entire rhombencephalon as reported in the ENCODE work of Gorkin et al., (2020). The rhombencephalon develops into the pontine nuclei, the medulla and other brainstem regions in addition to the cerebellum. Three points are notable here: (1) Our analysis, is specific to chromatin profiling of the developing cerebellum which is a novel dataset. (2) In our analysis, we examined the P9 timepoint which was not examined in the ENCODE dataset, and this is a key watershed in cerebellar development; at this timepoint we find a large subset of enhancers that are specific to postnatal cerebellar development. (3) When comparing our cerebellar histone profiles to those generated in the hindbrain, we find that ~25% of our H3K27ac peaks at E12 *are not* found in the ENCODE hindbrain data (Figure 2 —figure supplement 1A). Additionally, we identified 765 robust cerebellar enhancers active in the cerebellum and not in the hindbrain. Thus, our study has led to the discovery of novel enhancer elements, potentially important specifically for cerebellar development. These findings are presented in the Results section ‘A subset of cerebellar enhancers is active in other developing regions of the brain’ (pp. 10) in the revised manuscript. We also utilize our cerebellar histone profiles to uncover novel regulators of cerebellar development and characterize the expression and function for transcription factors Pax3 and Bhlhe22, which have not previously been described in the literature. In comparison to the ENCODE analysis previously conducted in the mouse hindbrain, our study provides results and datasets that are digestible and curated, which can be utilized to discover temporally-specific regulators of embryonic and postnatal cerebellar development.

While this focused study should have an audience, it would be stronger if the authors interfaced their data and results with other larger and more granular (e.g. single cell or more chromatin marks) datasets. If this is done effectively, this manuscript will have clear stand alone value as a complement to the larger, but not curated or cerebellum focused ENCODE data. And would link to the emergine single cell datasets that also focus on dynamic transcriptional and epigenomic changes. This would put the results from this work in the context of the larger and centralized efforts, while maintaining the useful focus.

We agree with this reviewer that the integration of the previously generated data will provide our study with stand-alone value as a complement to these larger datasets. We have addressed this reviewer’s suggest to interface our data with larger and more granular datasets (see our Cover Letter). Overall, by the addition of relevant comparisons, our study provides a valuable resource of active enhancers focused on the developing cerebellum and can be used as a means to discover novel genetic regulators of cerebellar development.

Analysis and results:1. The IHC in the manuscript lacks any quantitation and co-labeling/co-expression patterns are not clear with the lower magnification images. Figure 3 and 6 (and supp fig) analysis should have a quantitative summary of staining and of co-labeling. For example, Figure 3 is used to argue for presence or absence of co-expression/co-labeling between Pax3 and Ptf1a, Pax2, and Foxp2. I would like to see neuroanatomical annotation lines as overlay on sections for non-experts. Finally, I think higher magnification insets that clearly show co-labeling (or lack thereof) would make these figures stronger.

We agree with this reviewer that a quantitative assessment of co-labeling will add further value to our findings and that higher magnification images of co-staining experiments will bring more clarity to our results of our co-staining experiments. To these ends, in the revised manuscript, we have now included a quantitative analysis of co-staining relative to the protein of interest. For Pax3 we quantified the percentage of Pax3 cells that colocalized with Ptf1a, Pax2, Foxp2, and Calbindin. These changes for Pax3 are detailed in Figure 3 and Figure 3 —figure supplement 1. In the revised manuscript, this analysis can be found in the section ‘Early active enhancers are enriched for Pax3 binding sites, a novel marker for GABAergic cells.’ (pp. 14).

For Bhlhe22 and markers for differentiating granule cells and migrating cells, we quantified the percentage of Bhlhe22 cells that colocalized with Neurod1, NeuN and Dcx. These data for Bhlhe22 are shown in Figure 6—figure supplement 1. In the revised manuscript, this analysis can be found in the section ‘Bhlhe22 is a novel regulator of granule cell development’ (pp. 20-21).

Additionally, we ensure all the images used to assess immunofluorescent co-staining with cell-type specific markers were taken at 20X magnification. We have also drawn neuroanatomical annotation lines to identify regions of interest in the developing cerebellum.

2. The P0 enhancer set is much smaller than the E12 or P9, raising questions about technical sensitivity. The authors make the conclusion that there are no P0 enriched enhancers, but this could also simply be a lack of sensitivity. The authors could address this by presenting information regarding the signal to noise and comparing their data to published ENCODE P0 hindbrain data.

We thank the reviewer for this suggestion and in the revised manuscript we present quality control metrics recommended by ENCODE to evaluate the success and sensitivity of our H3K27ac and H3K4me1 immunoprecipitations: fraction of reads in peaks (FRiP) and relative strand correlation (RSC) (Landt et al., 2012). These metrics are explained in greater detail in Landt et al., 2012 and briefly in the reviewed Methods section. All the H3K27ac and H3K4me1 samples generated in our study met ENCODE standards for a successful immunoprecipitation, with FRiP score higher than 1% and an RSC value greater than 0.8 (Landt et al., 2012). The P0 samples had slightly lower scores, on average, in both metrics than the E12 and P9 samples, suggesting that the lower number of P0 enriched enhancers could have been due to a lack of sensitivity compared to the other stages. Additionally, the number of peaks identified in both biological replicates for H3K27ac was greater in the P0 ENCODE hindbrain compared to our cerebellar dataset, also indicating issues in sensitivity. However, 91.7% of the H3K27ac peaks identified in the cerebellum were replicated in the hindbrain, indicating robust signals. The comparisons between our dataset and the ENCODE hindbrain data are presented in the Results section ‘Enhancer dynamics during cerebellar development’ (pp. 8).

3. The authors should compare their datasets to the mouse ENCODE developmental hindbrain datasets. Ideally, they would verify the patterns they found, showing that their datasets are high quality and correlated with the ENCODE data and then also review the reproducibility of the early vs. late enhancers.

We thank the reviewer for this suggestion and agree that identifying whether our findings are aligned with the ENCODE hindbrain datasets would support our enhancer predictions and provide greater confidence that our datasets are of high quality. As noted in Response #1 to Reviewer #1, our study focused on enhancer activity that is specific to the developing cerebellum while ENCODE looked at enhancer activity throughout the hindbrain. As suggested by this Reviewer (2), we compared chromatin profiles generated in our study to those in the hindbrain and identified that the majority of H3K27ac and H3K4me1 peaks overlapped between the two datasets. There is a subset of peaks specific to the cerebellum (~25% of peaks H3K27ac peaks at each stage) that highlights the identification of cerebellarspecific enhancers, bolstering the import of our analysis. These findings are detailed in a new section of the manuscript entitled “A subset of robust cerebellar enhancers is active in other developing brain regions” (pp. 10) and in Figure 2 —figure supplement 1.

In the revised manuscript, we also assessed whether the activity patterns/histone ChIP-seq signal of Early and Late active enhancers could be reproduced in mouse hindbrain samples. We found that in the hindbrain, Early and Late enhancers showed similar trends as we find in the cerebellum with a slight increase and decrease in activity, respectively. These findings are detailed in the new section entitled ‘A subset of robust cerebellar enhancers is active in other developing regions of the brain’ (pp. 10) and in Figure 2—figure supplement 2 and 3.

The authors can also use the P9 data to compare to the adult cerebellum in mouse ENCODE to see how similar these datasets are – it would be of interest if P9 is capturing novel enhancers.

This is an interesting point and we agree that it is relevant to this study to identify whether cerebellar enhancers demonstrate postnatally specific activity compared to the adult. The postnatal day (P)9 cerebellum is far from developmentally static and previous postnatal findings by Frank et al., (2015) indicate that there may be enhancers with activity specific to postnatal development compared to adult stages. To examine this in our data, we have now included a comparison of mouse cerebellar P9 histone profiles with the ENCODE adult profiles. We find that 30-40% of peaks were unique to P9. This indicates that a subset of the enhancers identified in our study may have activity specific to postnatal development. These findings can be found in the revised Results section ‘Enhancer identification during cerebellar development’ (pp. 7).

The authors could also look at dynamic chromatin accessibility to see if the H3K27ac + H3K4me1 signature here is more sensitive to active enhancers than accessibility alone (which would be expected and would further justify the value of the data here).

In our revised manuscript, we address this comment by looking at the overlap of our predicted enhancers with experimentally validated enhancers in the VISTA enhancer database. We find that 0.05% H3K27ac peaks overlapped with experimentally validated enhancers with activity in the hindbrain. This compares very favorably with the same overlap of cerebellar snATAC-seq and hindbrain ATAC-seq peaks, where only 0.005% and 0.006% of peaks overlapped with VISTA enhancers, respectively. This comparison is detailed in the revised Results section ‘Enhancer identification during cerebellar development’ (pp. 7) and in Figure 1—figure supplement 1. This analysis further justifies the value of our data as a resource for the identification of regulatory sequences that are likely to be active enhancers in the cerebellum.

4. It would be of interest to see how the enhancer sets identified here map to the recently published snATAC-seq datasets from mouse cerebellum across development. This publication was referenced in the manuscript, but not described – Sarropoulos Science 2021. More specifically, can the early and late enhancers here be assigned to shared or distinct cerebellar cell types?

We thank the reviewer for this suggestion and is aligned with this revision’s status as a resource paper.

We now compare our cerebellar histone profiles to single-cell profiles of chromatin accessibility (Sarrapolous et al., 2021). Mapping our cerebellar enhancers with this recently published snATAC-seq dataset we find that 90.1% of robust cerebellar enhancers identified in our bulk approach overlapped with CREs identified using snATAC-seq. We then predicted the cell type in which our robust cerebellar enhancers were active based on the cell-type annotation of the overlapping CREs (see new Figure 2 —figure supplement 4). Specifically, as suggested by this Reviewer, we used this dataset to quantify Early and Late active robust cerebellar enhancer accessibility in distinct cell types. The comparison of our findings with the snATAC-seq dataset provides cell-type resolution of chromatin accessibility for the majority of cerebellar enhancers. Importantly, our dataset provides novel evidence of enhancer activity by identifying enhancer-associated histone marks at thousands of CREs identified by snATAC-seq. These sequences are more likely to function as enhancers and are prime candidates for validation. The analysis and results can be found in a new Results section in our revised manuscript entitled ‘Comparison of robust cerebellar enhancers with chromatin accessibility identified in single cells’ (pp. 11).

5. The authors could intersect their target early and late genes with recent snRNA-seq data from developing human cerebellum (Aldinger Nat Neuro 2021).

The suggestion made by this reviewer is excellent and in the revised manuscript we now quantify the expression of Pax3 and Bhlhe22 in various cell-types in the developing human cerebellum using the snRNA-seq dataset previously generated by Aldinger et al., (2021). Pax3 and Bhlhe22 are Early and Late genes, respectively, discovered by our analysis with novel expression patterns and functions in the context of the developing cerebellum. The results of this analysis support the spatial expression patterns identified in our immunofluorescent staining for both Pax3 and Bhlhe22. These findings are presented in the revised Results sections ‘Early active enhancers are enriched for Pax3 binding sites, a novel marker for GABAergic cells’ (pp. 14-15) for Pax3 and ‘Bhlhe22 is a novel regulator of granule cell development’ (pp. 21) for Bhlhe22.

6. Some general issues with the target gene set analsyis:. For the k-means clustering of target genes presented in Figures 4 and 5, the authors should provide rationale for using the k value selected.

We agree that the rationale for choosing the k-value should be more clearly detailed and in our revised manuscript we emphasize the details of the analysis conducted to determine the number of clusters used for k-means clustering. To arrive at the k value (the number of clusters) for our k-means clustering analysis of Early and Late target genes we conducted an Elbow Analysis to determine the k value with minimal intra-cluster variation [total within-cluster sum of square (WSS)]. This method calculates the WSS and plots WSS as a function of the number clusters. The WSS decreases as the number of k decreases and the optimal k-value is the point at which the WSS first starts to diminish and the differences as k increases are minimal. When plotted on a line graph, this point is visible as an elbow.

We determined the optimal number of clusters is 4 for both Early and Late active enhancers (Figure 4 —figure supplement 1A). These points are provided in the ‘Co-expressed putative target genes are expressed in spatially distinct areas of the developing cerebellum’ (pp. 15) section in the revised manuscript.

Finally I'm not sure what the GeneRatio measure is supposed to capture. IT would be more straightforward to use an observed vs. expected fold enrichment, as counts can skew depending on number of genes and fold enrichment is standard in GO analysis.

We thank the reviewer for pointing out the pitfalls of the gene ratio measure and we have changed the GO enrichment analysis to an observed vs. expected fold enrichment as suggested by the reviewer. In the revised manuscript, this was conducted for plots displaying the results of our analyses in Figure 4D and 5D. Additionally, we’ve changed the x-axis of Figure 5D to ‘-log10 adjusted p-value’ and terms were sorted based on significant enrichment (ascending adjusted p-value). We think that this improves our findings by emphasizing the most highly enriched GO terms in each set of target genes.

7. Motif analysis: Motifs are often similar for families, so motif to TF matching is imprecise at best. Fine to select best guess TF as authors do, but good to also show motif logo in Figure 3A and include mention that the actual TF could be a different family member unless orthogonal evidence exists.

In agreement with these comments concerning motif families and TF matching, we have revised the manuscript to recognize that the motifs enriched in our cerebellar enhancers may represent the binding of TF families with similar motifs, not just a single best match. To address this point, we now provide a supplementary table of our results and the protein families for the TF with the best matching motif. In addition, we have revised Figure 3A to include the enriched motif logos which provides context for the TF matching. Finally, in the corresponding text of the revised manuscript, we emphasize that the TF bound to an enhancer may potentially be a member of the same protein family, as they typically have similar DNA binding sites. We now provide a Table and text in the Results section ‘Cerebellar enhancers are enriched for neural transcription factor binding sites in an age-dependent manner’ (pp. 12-13) that addresses this point.

8. Intersection with ASD variants is flimsy. The authors should use some presumed negative for non-brain and other brain GWAS to show that the ASD SNP findings are relevant e.g. have stronger signal than presumed negative or less relevant phenotypes.

We thank the reviewer for this recommendation and we believe that the addition of a negative control using non-brain GWAS demonstrates that our ASD SNP enrichment is relevant. As a negative control, we assessed whether our active cerebellar enhancers were enriched for SNPs associated with chronic kidney disease. We indeed find that there is no significant enrichment of chronic kidney disease SNPs, in contrast to ASD SNPs which were enriched in our cerebellar enhancers. This new addition to our analysis can be found in the section ‘Active cerebellar enhancers are enriched for common and de novo genetic variants associated with autism spectrum disorder.’ (pp. 23)

The overall enrichment with de novo mutations in enhancesr does not demonistrate any specific cerebellar relevance unless these enhancers are not active in other brain regions. Other studies have showed general enrichment of de novo mutations in brain enhancers. Not sure whether the rare CNV with the cerebellar gene is useful or convincing of relevance as the N is 1 and the gene expression outside the cerebellum is bot shown or discussed. All this analysis is fine, but shouldn't be treated as a major finding. I think the other parts of this study are much stronger, so de-emphasizing this would not hurt the paper.

While we agree with the reviewer that our enrichment analysis was not conducted exclusively for enhancers with cerebellar-specific activity, we believe that there is value in assessing the enrichment of de novo mutations associated with neurodevelopmental disorders such as ASD. Identifying the collective consequences of mutations in enhancers active in multiple brain regions may lead to a better understanding of the complex presentation of disorders like ASD. In accordance with the reviewer’s suggestion, we emphasize in our revised manuscript that this analysis is meant to be an example of the relevance of our dataset to human development and ASD and demonstrate how it may be utilized to functionally annotate variation associated with neurodevelopmental disorders. We believe that it would be beyond the scope of the current study to delve further into the potential impacts of the CNVs on cerebellar development, however it sets the stage for future studies.

9. The argument that MGI mouse cerebellum phenotype is evidence for validity needs to be compared to some random set in order to evaluate how strong the 98/2261 finding is- e.g. via permutation analysis selecting random sets of 2261 genes to establish background.

We thank the reviewer for this suggestion and we have now conducted a permutation analysis to assess whether the identification of 98 cerebellar genes within our list of 2261 target genes (2261) is evidence of statistical enrichment of cerebellar function. To do so, we generated 10,000 permutations of 2261 genes randomly selected from all genes expressed in the cerebellum and assessing the number of cerebellar genes in each permutation. We found that cerebellar genes were indeed enriched in robust cerebellar enhancer target genes, with an adjusted p-value = 0.0405. This analysis in now included in the section ‘Co-expressed putative target genes are expressed in spatially distinct areas of the developing cerebellum’ (pp. 17) and in Figure 4—figure supplement 2.

Text1. The authors highlight their study as the first to characterize developmentally active enhancers, both in the abstract and discussion. This is an over-statement at best and the authors would do better to both remove this language and to directly deal with some of the other datasets, as described below.

We recognize that our study is not the first to characterize developmentally active enhancers and in line with the reviewers comment we now frame our work in the context of previous studies and emphasize that our study is best viewed as a valuable contribution to the catalog of active enhancers in the cerebellum. Our study provides a novel and unique dataset previously missing in the literature and can be used as a resource for the discovery of novel regulators of cerebellar development.

2. The introduction should include brief description of the other relevant studies, particularly the mouse ENCODE data and recent snATAC-seq and explain the novelty of approach here. The discussion should devote space to comparing the results from this study and the mouse ENCOE and snATAC-seq specifically.

We thank the reviewer for this suggestion and we now describe the relevant studies that have conducted assessments of enhancer activity in other brain regions (ENCODE, Gorkin et al., 2020) and chromatin accessibility at single cell resolution (snATAC-seq, Sarrapolous et al., 2021). We have included a brief description of these studies in the Introduction (pp. 3-4) of the revised manuscript. This emphasizes the novelty of our approach and results. We also provide a discussion on the comparison of our findings with the ENCODE and snATAC-seq datasets.

Reviewer #3:Weaknesses:The authors select three timepoints during cerebellar development for their ChIP-seq analysis. The rational for selecting these specific timepoints is not very clear, especially as they do not cover the entire period of cerebellar development in the mouse.

We address the reviewer’s concern about the rationale for investigating the time periods chosen for ChIP-seq in the revised version of our manuscript by including additional text in the section ‘Enhancer identification during cerebellar development’ (pp. 5) in our results, noting the importance of these periods for the various stages of development across the principle cell types of the cerebellum. Thus, our study is a novel genome-wide assessment of enhancer-associated histone marks throughout key points in the developing cerebellum.

While the resource itself is very valuable, the presentation of example genes that the authors extract from their analyses (Pax3, Bhlhe22, and Cdc42bpb) is rather short and lacking in detail. For example, the authors could have further strengthened their findings by utilising recent single-cell datasets, in particular the single-cell transcriptional atlas of the developing cerebellum published by Carter et al.

This an important point made by the reviewer, particularly in the context of this being a resource paper where the value lies not only in our initial analysis but in it’s utility when combined with other existing datasets. We have strengthened our analyses by comparing our data with previous scRNA-seq datasets quantified in the developing mouse (Carter et al., 2018) and human (Aldinger et al., 2021) cerebellum. Comparisons were conducted specifically for the genes with novel functions in the context of cerebellar development Pax3 and Bhlhe22, where we examine their expression in the cell-types of the cerebellum. We also take the initial steps in validating these findings through immunofluorescent staining at several stages of development and co-staining with cell-type specific markers. We demonstrate the utility of our data as a resource and provide a proper framework for investigating genes with novel functions in the cerebellum. These analyses can be found in sections ‘Early active enhancers are enriched for Pax3 binding sites, a novel marker for GABAergic cells’ (pp. 14-15) and ‘Bhlhe22 is a novel regulator of granule cell development’ pp (21).

The authors use in silico tools to identify transcription factor binding sites enriched in the identified active enhancers. No experimental validation is provided for this analysis. This would have been particularly relevant for transcription factors that have not been previously well-described in cerebellar development including Pax3, which the authors present as a novel marker for GABAergic progenitors.

Based on the recommendations by the Editors of this study and the fact that the corresponding data will now be available as a Resource article, we believe conducting the experimental validation of transcription factor binding is outside the scope of this revision. The goal of our motif enrichment analysis is to predict the regulators of cerebellar enhancer activity. The identification of these TFs provides further insight into the cell-types in which these enhancers may be active and may help predict the cellular processes regulated by cerebellar enhancers. We do demonstrate how our predictions can be validated through a preliminary set of staining experiments for Pax3, a TF enriched in our cerebellar enhancers with unknown function in the cerebellum. This sets the stage for further investigation of Pax3 function, as well as for other novel regulators of cerebellar development identified by our analyses.

Furthermore, it would have been helpful to go into further detail and to provide some predicted candidate genes that might be regulated by Pax3 in GABAergic progenitors.

We thank the reviewer for this suggestion and agree that providing further detail and characterizing the enhancers with Pax3 motifs and their putative target genes may provide a more detailed description of Pax3 function in cerebellar development. To bolster our Pax3 findings, we identified the cerebellar enhancers with a Pax3 motif and their target genes, as per the reviewer’s suggestion. We found that 923 Early active robust cerebellar enhancers with a Pax3 binding motif were highly correlated with a putative target gene. A GO enrichment analysis determined that predicted Pax3 target genes were enriched for biological processes such as ‘regionalization’, ‘pattern specification process’ and ‘neural precursor cell proliferation’ which is in line with the results of our immunofluorescent analysis.

Additionally, we identified several target genes associated with the function of neural progenitors in the cerebellar VZ (ventricular zone) and the specification and differentiation of GABAergic cell types such as Ascl1, Neurog2, Sox11 and Ctnna2 (Figure 4 —figure supplement 2). These findings are detailed in the section ‘Co-expressed putative target genes are expressed in spatially distinct areas of the developing cerebellum’ (pp. 17).

Figure 3: Very limited expression data is shown for Pax3. It would have been helpful to show additional immunostainings at least for P9 (same timepoint as used for the ChIP-seq analysis) if not for further timepoints during cerebellar development. Moreover, it would have been really interesting to compare this to the data available on Pax3 from Carter et al. The authors mention this in their discussion; it would be helpful to include this data in the relevant figure and also to comment on how expression of Pax3 changes during the course of cerebellar development.

In agreement with the reviewer’s suggestions, we have extended the time course of Pax3 immunofluorescent staining to P3 and P9. We found that expression patterns found in P0 extend to P3 but by P9 we cannot detect Pax3 expression. These findings can be found in Figure 3 and Figure 3—figure supplement 1.

I am not sure that I agree with the authors about the fact that 98 out of 2261 identified target genes result in a cerebellar phenotype when knocked out particularly demonstrates the validity of the high throughput approach (page 12). Is this a significant number?

Addressed. See response to Reviewer #2 (Recommendations for the authors) Point 12.

The use of CAGE-sequencing data in this manuscript is very unclear. The authors refer to a previous manuscript (Zhang et al., 2018); however, in this study reports transcriptomic data from only three timepoints (E13, E15, E18) and not 12 timepoints as mentioned in by the authors? The authors should better describe which data they used. Also, given that the CAGE-Seq data is from bulk-sequencing, it would have been much more informative to look at gene expression of their candidate transcription factors in the Carter et al., single-cell dataset. Furthermore, this would have allowed the authors to look beyond the P9 timepoint and thus for a more complete gene expression analysis over cerebellar development. This is also true for Figures 1E,G.

We thank the reviewer for bringing up the CAGE data issue as confusing. This confusion was engendered by the use of an incorrect citation (Zhang et al., 2018). Our apologies, the correct reference is: Ha, T. J., Zhang, P. G. Y., Robert, R., Yeung, J., Swanson, D. J., Mathelier, A.,... Goldowitz, D. (2019). Identification of novel cerebellar developmental transcriptional regulators with motif activity analysis. BMC Genomics, 20(1), 718. doi:10.1186/s12864-019-6063-9. This has been corrected.

The approach of using the enhancer data set to identify novel target genes not previously identified in cerebellar development is unclear to me. Using their described filtering approach, the authors describe genes that differentially regulated by Atoh1. Could these not have been identified from the original dataset by Klisch et al., 2001? What does the initial identification of the enhancers add here?

The Klisch et al., (2001) dataset produced a large number of differentially expressed genes (DEGs) at P7. While ground-breaking in their analysis, it left two big gaps: (1) other time points were not assessed where many developmental events are occurring and (2) this produced about 2000 DEGs making it difficult to identify specific novel gene candidates that are critical for the postnatal differentiation of granule cells. To address these gaps, we (1) analyze enhancer activity and gene expression over embryonic and postnatal development (E12, P0, P9), and (2) due to our analysis of enhancers, which are likely to be regulating genes important for specification and differentiation, this allowed us discover novel TFs not previously identified in cerebellar development. We use our enhancer putative target gene dataset in conjunction with DEGs in the Atoh1 mouse to identify Bhlhe22, a novel TF that we find is important for postnatal granule cell migration. This analysis can be found in the section of the Results entitled ‘Bhlhe22 is a regulator of postnatal granule cell development’ (pp. 18).

Figure 6: As above, the authors should include analysis of Bhlhe22 expression in the Carter et al., single-cell dataset.

Addressed.

The authors normalize gene expression to co-transfected GFP. Can the authors demonstrate that cells that are transfected with GFP were also transfected with the siRNA construct?

We thank the reviewer for the suggestions and the aesthetic adjustments were made by increasing the font size of Figure 7 and bringing greater clarity to our results. We have provided quantitative RT-qPCR data of the reduction of Bhlhe22 in our siRNA treated cultures as evidence of a successful knockdown of Bhlhe22, and believe that immunostaining is not necessary in this regard. Co-transfection with a GFP plasmid and siRNA molecules has been previously been demonstrated as a means for identifying transfected cells in primary neuronal cultures (Tong et al., 2011; Wang et al., 2013; Schmidt et al., 2009; Cosker et al., 2008; Tonges et al., 2011). The co-transfection efficiency of this method has been demonstrated in human embryonic stem cells by Moore et al., 2010 (DOI:10.1186/scrt23), where siRNA molecules targeting GFP were co-transfected with a GFP plasmid by nucleofection, which is the transfection method used in our study. Fluorescence microscopy showed that GFP expression was visibly reduced when GFP-targeting siRNA was co-transfected FACS analysis showed that an siRNA to DNA ratio of 3:1 reduced the percentage of cells expressing GFP by 71.75% when compared to controls indicating high co-transfection efficiency. This has also been demonstrated in nucleofection protocol documentation provided by Lonza, which is the is manufacturer of the Amaxa Nucleofector 2b which is used in our study for transfection (https://bioscience.lonza.com/lonza_bs/US/en/document/21126). In this example, three different cell lines are transfected with GFP plasmid and GFP targeting siRNA. A reduction in GFP expression was observed in all cell lines, indicating high co-transfection efficiency. The co-transfection of a GFP plasmid with siRNA molecules is now common practice in the field, and that cells that are successfully electroporated with GFP typically are also transfected with the siRNA molecules.

Can the authors comment on the evolutionary conservation of their identified enhancer elements? This is particularly relevant for their analysis of variants associated with ASD. Are the loci of interest conserved between mouse and human and thus, do enhancer activity and transcription factor occupancy occur at orthologous regions distal to the same gene promoters?

All of the enhancers analyzed in our enrichment analysis of ASD variation are by definition conserved between mouse and human since the relevant human sequences used for this analysis were converted from the mouse to human genome using the UCSC Genome Browser LiftOver tool. This computational tool is called LiftOver because it converts coordinates in a reference assembly to another genome build. This is commonly used to perform cross species mapping, which takes coordinates of sequences in one species to identify the corresponding coordinates in other species. The majority of robust cerebellar enhancers (6630/7024) were converted from the mouse genome (mm9) to the human genome (hg38). 89.6% of robust cerebellar enhancers with a putative target gene are located at orthologous regions distal to the same gene promoters indicating conservation between mouse and humans and bringing credence to our analysis. The results of this analysis can be found in Supplementary Table 4 and in the revised manuscript in the section ‘Active cerebellar enhancers are enriched for common and de novo genetic variants associated with autism spectrum disorder.’ (pp. 23)

For the identification of CDC42BPB, it would have been nice to link this back to expression data over time and to provide more detail on the potential transcription factors involved.

We thank the reviewer for this suggestion and agree that it is relevant to our study to examine CDC42BPB expression in the developing mouse expression time course. In our revised manuscript, we describe CDC42BPB expression throughout embryonic and postnatal development. We also profile expression in the cell-types of the cerebellum using scRNA-seq in mouse and human samples (Carter et al., 2018; Aldinger et al., 2021). Finally, we confirm that the enhancer predicted to regulate CDC42BPB is associated with the same gene in the mouse and located on the same chromosome and at a similar distal genomic location in the human. (pp. 25)

Finally, given that this is a very important dataset for the scientific community, I wish it would be easier for others to interact with the data rather than just depositing data files into GEO. It would have also been really helpful to provide supplementary tables listing enhancer elements and associated transcription factors as well as nearby genes that the authors have identified. Currently, it is not transparent which lists of genes etc were used for the various analyses shown.

We agree with the reviewer that this is a very important dataset for the scientific community and have made efforts to make our data more accessible. An online resource

(https://goldowitzlab.shinyapps.io/developing_mouse_cerebellum_enhancer_atlas/) was created that can be used to browse, curate and export information regarding cerebellar enhancers and their putative target genes identified in this study. In the revised manuscript, we have also provided additional supplementary tables listing enhancer elements, their associated transcription factors and their associated target genes.